# Sensing of viral and endogenous RNA by ZBP1/DAI induces necroptosis

Jonathan Maelfait[1,†,‡,]*, Layal Liverpool[1], Anne Bridgeman[1], Katherine B Ragan[2],
Jason W Upton[2] & Jan Rehwinkel[1,]**

## Abstract

Nucleic acids are potent triggers for innate immunity. Double-stranded DNA and RNA adopt different helical conformations, including the unusual Z-conformation. Z-DNA/RNA is recognised by Z-binding domains (ZBDs), which are present in proteins implicated in antiviral immunity. These include ZBP1 (also known as DAI or DLM-1), which induces necroptosis, an inflammatory form of cell death. Using reconstitution and knock-in models, we report that mutation of key amino acids involved in Z-DNA/RNA binding in ZBP1's ZBDs prevented necroptosis upon infection with mouse cytomegalovirus. Induction of cell death was cell autonomous and required RNA synthesis but not viral DNA replication. Accordingly, ZBP1 directly bound to RNA via its ZBDs. Intact ZBP1-ZBDs were also required for necroptosis triggered by ectopic expression of ZBP1 and caspase blockade, and ZBP1 cross-linked to endogenous RNA. These observations show that Z-RNA may constitute a molecular pattern that induces inflammatory cell death upon sensing by ZBP1.

**Keywords** ZBP1; DAI; Z-RNA; cytomegalovirus; necroptosis
**Subject Categories** Immunology; RNA Biology
**The EMBO Journal (2017) 36: 2529–2543**

See also: **S Assil & SR Paludan** (September 2017)

## Introduction

Nucleic acids form double-stranded helices. Double-stranded (ds) DNA typically adopts the so-called B-conformation, while dsRNA is usually in the A-conformation. Both DNA and RNA can also adopt a Z-form double helix that is characterised by a left-handed helical arrangement, a zigzag pattern of the phosphodiester backbone and the absence of major grooves (Wang *et al*, 1979; Hall *et al*, 1984; Rich & Zhang, 2003). Antibodies raised against Z-form RNA or DNA stain mammalian cells and chromosomes, respectively, suggesting that nucleic acids in

this unusual conformation occur naturally in cells (Viegas-Péquignot *et al*, 1983; Zarling *et al*, 1990). However, biological functions of the Z-conformation are only partially understood (Rich & Zhang, 2003; Wang & Vasquez, 2007). Z-DNA is believed to play a role in transcription by relieving torsional strain induced within the DNA template by the movement of RNA polymerases, and Z-DNA may also promote genetic instability (Rich & Zhang, 2003; Wang & Vasquez, 2007). Physiological functions of Z-RNA remain unknown.

Early biochemical and structural studies identified a protein fold called Z-binding domain (ZBD) that specifically binds to Z-DNA but not to B-DNA (Herbert *et al*, 1993; Schwartz *et al*, 1999). This 78-amino-acid domain is also referred to as Zα or Zβ, and subsequent work showed that it not only binds to Z-DNA but also to Z-RNA (Brown *et al*, 2000; Placido *et al*, 2007). ZBDs have been identified in four proteins: ADAR1, ZBP1, E3L and PKZ (Athanasiadis, 2012). ADAR1 and ZBP1 are mammalian proteins implicated in antiviral innate immune responses. E3L is a poxvirus protein known to antagonise this host response. Lastly, PKZ is a fish protein related to mammalian PKR, also involved in antiviral immunity. This suggests an important role of ZBDs in innate antiviral immune responses (Athanasiadis, 2012) and may imply that Z-DNA and/or Z-RNA trigger such a host defence response. Z-form nucleic acids are further implicated in immune responses by the observation that sera from some systemic lupus erythematosus patients contain anti-Z-DNA autoantibodies (Lafer *et al*, 1983).

Z-DNA binding protein 1 (ZBP1, also known as DAI or DLM-1) has two N-terminal ZBDs (Zα1 and Zα2) and two RIP homotypic interaction motifs (RHIMs) (Ha *et al*, 2008; Kaiser *et al*, 2008; Rebsamen *et al*, 2009; Kim *et al*, 2011a,b; Yang *et al*, 2014). ZBP1 had initially been suggested to be a cytosolic B-DNA sensor that induces type I interferons (IFNs) (Takaoka *et al*, 2007). However, this observation could not be confirmed in ZBP1-deficient mice (Ishii *et al*, 2008). Instead, ZBP1 activates NF-κB and triggers necroptosis (Kaiser *et al*, 2008; Rebsamen *et al*, 2009; Upton *et al*, 2012), an inflammatory form of programmed cell death (Wallach *et al*, 2016). NF-κB activation is mediated by RIPK1 and RIPK3 kinases, which bind to ZBP1 via RHIM-RHIM interactions (Kaiser *et al*, 2008; Rebsamen *et al*, 2009). In contrast, ZBP1-dependent

1 Medical Research Council Human Immunology Unit, Radcliffe Department of Medicine, Medical Research Council Weatherall Institute of Molecular Medicine, University of Oxford, Oxford, UK
2 Department of Molecular Biosciences, LaMontagne Center for Infectious Disease, Institute for Cellular and Molecular Biology, University of Texas at Austin, Austin, TX, USA
*Corresponding author. Tel: +32 9 331 3745; E-mail: jonathan.maelfait@irc.vib-ugent.be
**Corresponding author. Tel: +44 1865 222362; E-mail: jan.rehwinkel@imm.ox.ac.uk
†Present address: Laboratory of Immunoregulation and Mucosal Immunology, VIB Center for Inflammation Research, Ghent, Belgium
‡Present address: Department of Internal Medicine, Ghent University, Ghent, Belgium

necroptosis depends on RIPK3 recruitment and does not require RIPK1 (Upton *et al*, 2012). RIPK3 phosphorylates and activates the pseudokinase MLKL, which upon oligomerisation ruptures the plasma membrane, leading to cell death (Wallach *et al*, 2016).

Necroptosis mediates host defence by eliminating virus-infected cells (Mocarski *et al*, 2015). Viruses in turn suppress necroptosis. A well-studied example is mouse cytomegalovirus (MCMV) infection (Mocarski *et al*, 2015): MCMV prevents ZBP1- and RIPK3-dependent necroptosis by expression of M45 (Upton *et al*, 2010, 2012). M45 has a RHIM that disrupts the interaction of ZBP1 and RIPK3. MCMV expressing M45 with a mutated RHIM fails to inhibit necroptosis, resulting in death of infected cells (Upton *et al*, 2012). Necroptotic cell death is normally held in check by the proteolytic activity of the pro-apoptotic caspase-8. Cells can be sensitised to RIPK3-dependent necroptosis by genetic loss (Kaiser *et al*, 2011; Oberst *et al*, 2011) or chemical blockade of caspase-8 (Vandenabeele *et al*, 2006). During M45-RHIM-mutant MCMV infection, the viral M36 protein—also known as viral inhibitor of caspase-8-induced apoptosis (vICA)—is believed to predispose cells to necroptosis by blocking caspase-8 (McCormick *et al*, 2003; Ménard *et al*, 2003; Guo *et al*, 2015).

Here, we show that necroptosis triggered by MCMV-M45^mutRHIM or by ectopic ZBP1 expression and caspase blockade required binding of nucleic acids to the tandem ZBDs of ZBP1. We generated *Zbp1* knock-in mice carrying four amino acid substitutions that abrogate binding to Z-form nucleic acids. Cells from *Zbp1*-mutant mice are resistant to MCMV-M45^mutRHIM-induced necroptosis. During infection, inhibition of RNA transcription but not of viral DNA replication prevented ZBP1-dependent necroptosis and we found that ZBP1 bound to newly synthesised RNA. Furthermore, ZBP1 cross-linked to endogenous RNA and stained cells in an RNase-sensitive manner. In sum, these results show that ZBP1 is an RNA sensor.

## Results

### Cell death following MCMV-M45^mutRHIM infection requires intact ZBP1 ZBDs

We initially wished to confirm the role of ZBP1 in cell death upon virus infection. Expression of ZBP1 was induced by IFN in *Zbp1*^+/−

but not in *Zbp1*^−/− primary mouse embryonic fibroblasts (MEFs; Fig 1A). We then assessed cell viability upon virus infection by determining intracellular ATP levels using CellTiter-Glo reagent. Infection with MCMV-M45^mutRHIM, a virus that fails to inhibit ZBP1-RIPK3-dependent necroptosis (Upton *et al*, 2012), induced cell death in *Zbp1*^+/− primary MEFs following IFN pre-treatment (Fig 1B). In contrast, viability upon MCMV-M45^mutRHIM infection was much higher in IFN-treated *Zbp1*^−/− primary MEFs (Fig 1B), despite comparable expression of RIPK3 and MLKL (Fig 1A). We also infected primary MEFs with MCMV-M45^wt, which did not cause much cell death in either *Zbp1*^+/− or *Zbp1*^−/− cells (Fig 1B). As a control, the induction of necroptosis by TNF stimulation in the presence of the caspase inhibitor zVAD (Vercammen *et al*, 1998) (hereafter simply TZ) was independent of ZBP1 (Fig 1B). Similar results were obtained in immortalised MEFs (Fig EV1A and B). These observations are consistent with an earlier report (Upton *et al*, 2012) and demonstrate that death of MCMV-infected cells required ZBP1 and was antagonised by the viral M45 protein.

To test if binding of Z-form nucleic acids to ZBP1 is involved in the induction of cell death during virus infection, we mutated key conserved residues involved in Z-RNA/DNA binding in mouse ZBP1. Based on the structures of the ZBP1 Zα1 and Zα2 domains (Schwartz *et al*, 2001; Ha *et al*, 2008; Kim *et al*, 2011a,b; Yang *et al*, 2014) and on sequence alignments (Fig 1C and D), we introduced N46A and Y50A mutations into Zα1 (Zα1^mut), N122A and Y126A substitutions into Zα2 (Zα2^mut), and all four mutations together (Zα1α2^mut). Mutation of the corresponding residues in ADAR1's Zα ZBD completely abolishes Z-DNA binding without altering protein stability (Schade *et al*, 1999); accordingly, we predicted diminished binding of our ZBP1 mutants to Z-RNA/DNA. Wild-type and Zα1α2^mut ZBP1 proteins were equally able to induce an NF-κB reporter upon ectopic expression together with RIPK3 in HEK293T cells, suggesting that the mutant protein folds correctly and retains signalling via its RHIMs (Figs 1E and EV1C).

We reconstituted immortalised *Zbp1*^−/− MEFs with ZBP1 3xFLAG-tagged ZBD mutants (Fig 1F). Cell viability following TZ treatment or MCMV-M45^wt infection was comparable between parental *Zbp1*^−/− and ZBP1-reconstituted cells (Fig 1G). MCMV-M45^mutRHIM infection induced pronounced cell death in ZBP1 wild-type reconstituted cells (Fig 1G and H). In contrast, ZBP1-Zα1α2^mut-expressing cells displayed higher viability after infection and

**Figure 1.  Necroptosis during MCMV infection requires nucleic acid binding to the ZBDs of ZBP1.**

A, B  Primary MEFs of the indicated genotypes were treated or not with 100 U/ml of IFN-A/D for 16 h. (A) Cell extracts were subjected to Western blot analysis using the indicated antibodies. (B) Cells were treated with 30 ng/ml TNF and 20 μM zVAD (TZ) or were infected with the indicated viruses at an MOI of 10. After 16 h, cell viability was assessed using CellTiter-Glo reagent. Values for untreated cells were set to 100%.

C  Domain architecture of mouse ZBP1.

D  Conservation of key amino acids in ZBP1's ZBDs. ZBP1 sequences from the indicated species were aligned using Clustal Omega and the sequence context of the conserved asparagine and tyrosine residues involved in Z-DNA/RNA binding is shown.

E  HEK293T cells were transfected with 50 ng NF-κB firefly luciferase and 25 ng *Renilla* luciferase reporter plasmids, together with expression vectors for RIPK3 (0.2 ng) and HA-STING or ZBP1-3xFLAG (20, 100, 500 ng). Luciferase activity was measured after 24 h, and the ratio of firefly and *Renilla* luciferase was set to 1 for control cells transfected with empty vector. Cell lysates were analysed for expression of the indicated proteins by Western blot (bottom). Asterisk (*) indicates residual signal from the α-HA antibody.

F–I  Immortalised *Zbp1*^−/− MEFs were reconstituted with the indicated ZBP1 mutants. (F) Cell extracts were subjected to Western blot analysis using the indicated antibodies. (G and H) Cell viability upon TZ treatment or MCMV infection was assessed as in (B). In (G), an MOI of 10 was used. (I) Cell death was monitored upon infection (MOI = 3) or TZ treatment using an in-incubator imaging platform (Incucyte) and the dye Sytox Green that stains dead cells.

Data information: Data are representative of three or more independent experiments. Panels (A, B, E and G–I) represent mean ± SD (*n* = 3). **P < 0.01, ***P < 0.001; two-way ANOVA. See also Fig EV1.

Source data are available online for this figure.

behaved similar to the parental $Zbp1^{-/-}$ cells, while ZBP1-Zα1$^{mut}$ and ZBP1-Zα2$^{mut}$ cells showed intermediate phenotypes (Fig 1G and H), with a greater contribution to cell death of the Zα2 domain. Cell death induced by MCMV-M45$^{mutRHIM}$ in wild-type ZBP1-expressing cells was also observed by live-cell microscopy using the dye Sytox-Green, which stains cells that have lost membrane integrity, and cell death was detected from 8 h post-infection onwards (Fig 1I). We then used mouse NIH3T3 fibroblasts, a transformed cell line.

Endogenous ZBP1 expression was undetectable by Western blot, but could be induced by IFN treatment (Fig EV1D). We stably transduced NIH3T3 cells to express either 3xFLAG-tagged wild-type or mutant ZBP1 (Fig EV1D and F). Cell viability was comparable between these cells after MCMV-M45$^{wt}$ infection (Fig EV1E) and upon TZ treatment (Fig EV1G and H). Consistent with our results in MEFs, cell viability was decreased in wild-type ZBP1 but not in ZBP1-Zα1α2$^{mut}$-expressing cells following infection with

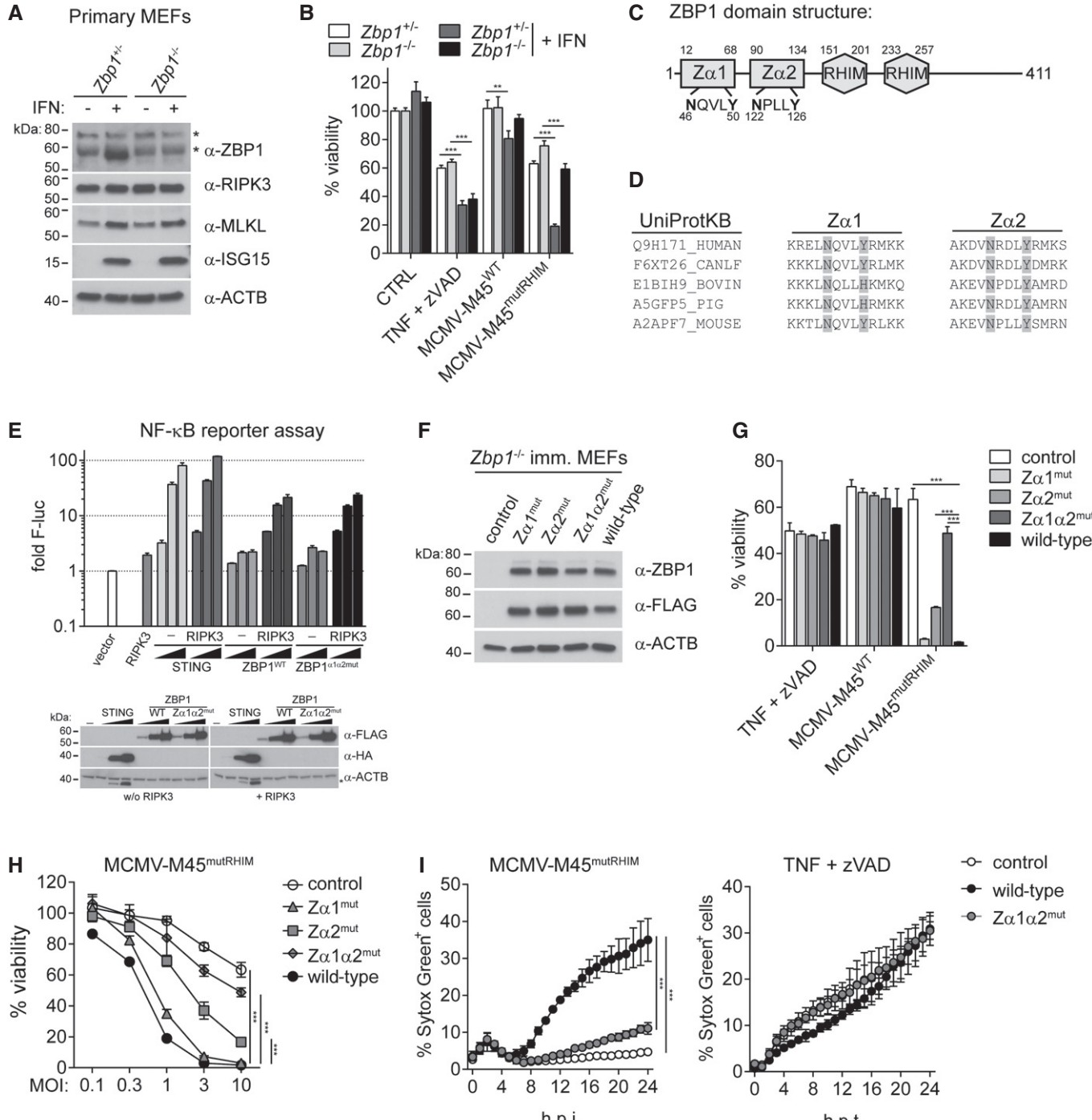

**Figure 1.**

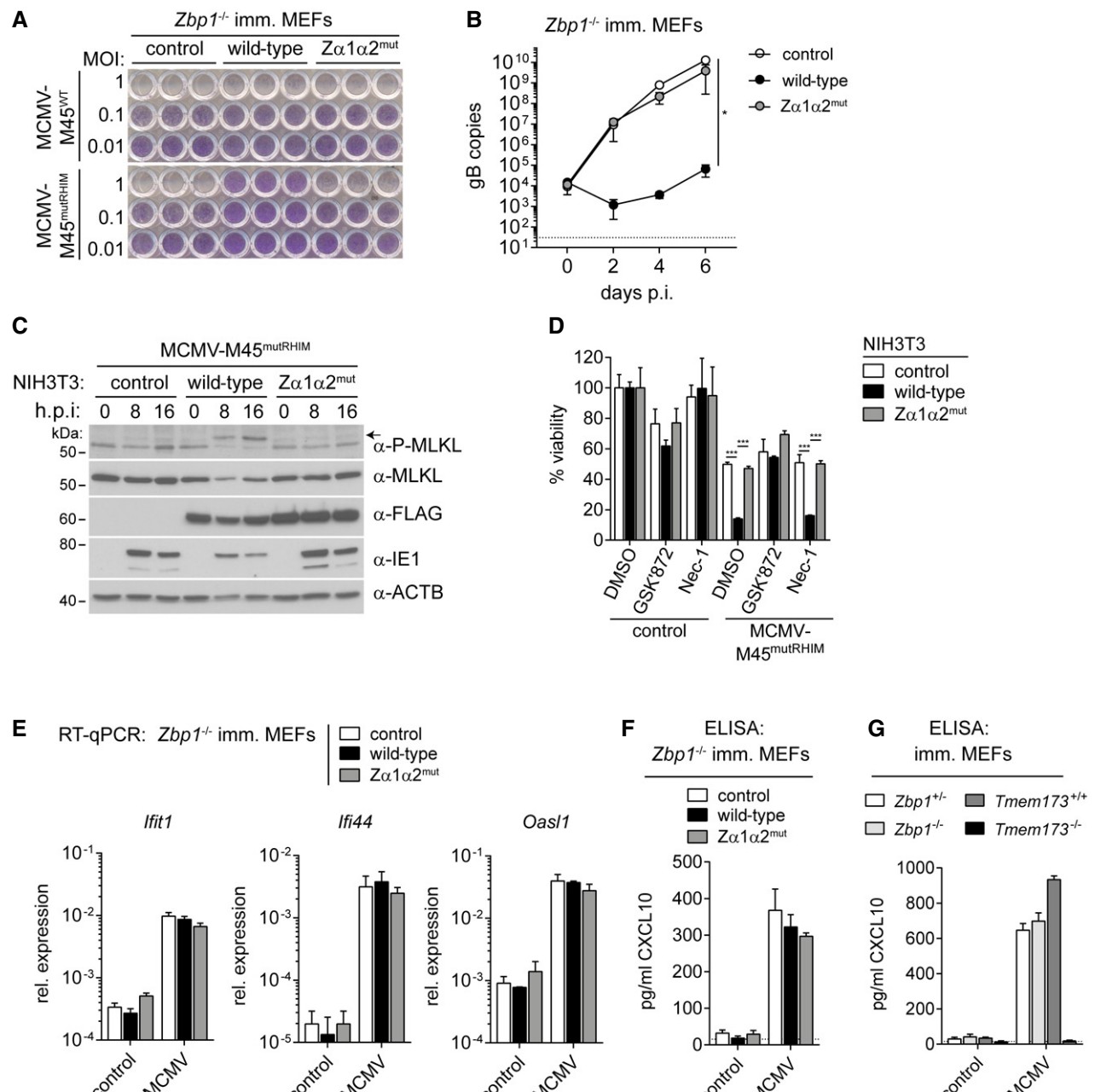

**Figure 2. ZBP1 restricts MCMV replication by inducing necroptosis but not IFN expression.**

A    Immortalised *Zbp1*[−/−] MEFs reconstituted with wild-type or mutant ZBP1 were infected as indicated. After 5 days, cells were fixed and stained with crystal violet.

B    Immortalised *Zbp1*[−/−] MEFs reconstituted with wild-type or mutant ZBP1 were infected with MCMV-M45[mutRHIM] at an MOI of 0.01. DNA samples were collected at 0, 2, 4 or 6 days after infection and analysed by qPCR. gB copy numbers were derived from a titration using defined amounts of gB plasmid. The dotted line represents the lower limit of detection.

C    NIH3T3 cells reconstituted with wild-type or mutant ZBP1 were infected as indicated using an MOI of 3. After 0, 8 or 16 h, cell lysates were subjected to Western blot analysis under reducing conditions using the indicated antibodies. The arrow indicates phosphorylated MLKL.

D    NIH3T3 cells reconstituted with wild-type or mutant ZBP1 were infected with MCMV-M45[mutRHIM] at an MOI of 10. Cells were simultaneously treated with 3 μM GSK'872 (RIPK3 inhibitor) or 30 μM Nec-1 (RIPK1 inhibitor) or 0.1% DMSO as a control. Cell viability was analysed 24 h after infection as in Fig 1B.

E    Immortalised *Zbp1*[−/−] MEFs reconstituted with wild-type or mutant ZBP1 were infected with MCMV-M45[mutRHIM] at an MOI of 3. RNA samples were collected 8 h after infection and analysed by RT–qPCR.

F, G    CXCL10 was analysed by ELISA in supernatants from MEFs of the indicated genotypes infected with MCMV-M45[mutRHIM] (MOI = 3; 8 h post-infection). The dotted line represents the lower limit of detection.

Data information: Data are representative of two or more independent experiments. Panels (B and D–G) represent mean ± SD (n = 3). *P < 0.05, ***P < 0.001; two-way ANOVA. See also Fig EV2.

Source data are available online for this figure.

MCMV-M45$^{mutRHIM}$ (Fig EV1E–H). Taken together, these observations suggest that induction of cell death by ZBP1 requires intact ZBDs and may imply that nucleic acids in Z-conformation trigger this response.

## Nucleic acid recognition by ZBP1 is essential for necroptosis and virus restriction, but not for IFN induction

We next wished to test whether restriction of MCMV infection by ZBP1 depends on the protein's ZBDs. Infection of cells with MCMV results in lytic cell death after several days of culture due to virus replication. Expression of wild-type ZBP1 but not ZBP1-Zα1α2$^{mut}$ protected $Zbp1^{-/-}$ MEFs or NIH3T3 cells against MCMV-M45$^{mutRHIM}$-induced cytopathic effects 5 days post-infection (Figs 2A and EV2A). Furthermore, replication of the virus genome was blunted by wild-type but not by mutant ZBP1 (Fig 2B). These observations show that ZBP1 defended cells against MCMV and that intact ZBP1 ZBDs were required for this.

This block in virus replication may depend on ZBP1-dependent cell death or engagement of an IFN response in infected cells. We first explored how mutations in ZBP1's ZBDs affected MCMV-M45$^{mutRHIM}$-induced cell death. Previous work suggested that the type of cell death triggered by ZBP1 upon MCMV-M45$^{mutRHIM}$ infection is necroptosis (Upton *et al*, 2012). Upon phosphorylation by RIPK3, MLKL forms trimers and large multimeric complexes, which execute necroptosis by causing plasma membrane permeabilisation (Wallach *et al*, 2016). Phosphorylation of MLKL was only observed in MCMV-M45$^{mutRHIM}$-infected NIH3T3 cells reconstituted with wild-type ZBP1 and not in ZBP1-Zα1α2$^{mut}$ ZBP1-expressing cells (Fig 2C). Concomitantly, reduced levels of the viral IE1 protein accumulated in cells expressing wild-type ZBP1 (Fig 2C). Furthermore, we observed the formation of MLKL-trimers (Cai *et al*, 2014) in non-reducing SDS–PAGE gels after induction of necroptosis in NIH3T3 cells with TZ (Fig EV2B). Large MLKL oligomers, which did not migrate through the gel, were also observed in the gel-loading pocket (Dondelinger *et al*, 2014; Fig EV2B). MCMV-M45$^{wt}$ infection did not change MLKL electrophoretic mobility (Fig EV2B), which is in line with our earlier observation that this virus does not cause cell death (Fig 1). MCMV-M45$^{mutRHIM}$ infection induced MLKL oligomerisation at 16 h post-infection (Figs 2C and EV2B). These effects were enhanced and accelerated in cells expressing wild-type ZBP1 but not in cells reconstituted with ZBP1-Zα1α2$^{mut}$ (Fig EV2B). Furthermore, the viability of MCMV-M45$^{mutRHIM}$-infected NIH3T3 cells expressing wild-type ZBP1 was rescued upon treatment with the RIPK3 kinase inhibitor GSK'872 (Fig 2D). As described previously (Upton *et al*, 2010), blocking RIPK1 activity with Nec-1 had no effect on virus-induced cell death (Fig 2D). These observations show that ZBP1 indeed activates necroptosis in this setting.

MCMV-M45$^{mutRHIM}$ infection also induced an IFN response. Expression of the interferon-stimulated genes (ISGs) *Ifit1*, *Ifi44* and *Oasl1* was increased upon infection and was not altered in cells expressing ZBP1 (Fig 2E). Consistent with this observation, the expression and secretion of CXCL10, a chemokine that signifies the induction of an IFN response, were independent of ZBP1 or MCMV M45 protein (Figs 2F and EV2C). Instead, CXCL10 induction was reduced to background levels in $Tmem173^{-/-}$ cells that lack STING, a protein involved in cytosolic DNA sensing (Figs 2G and EV2C; Chen *et al*, 2016). Furthermore, overexpression of ZBP1 did not

induce an IFNβ reporter in HEK293T cells (Fig EV2D). In sum, these results demonstrate that ZBP1 and its ZBDs were required specifically for the induction of necroptosis upon virus infection and not for the induction of an IFN response.

## ZBP1-Zα1α2$^{mut}$ knock-in mice do not restrict MCMV-M45$^{mutRHIM}$ and cells from these animals do not undergo virus-triggered necroptosis

To substantiate our conclusion that intact ZBDs are required for ZBP1-dependent necroptosis, we developed a knock-in mouse in which the endogenous *Zbp1* allele is replaced by *Zbp1-Zα1α2$^{mut}$* (*Zbp1$^{Zα1α2}$*) carrying the four point mutations shown in Fig 1C. The generation and validation of this model are detailed in Fig EV3A and B. We obtained primary MEFs by crossing of $Zbp1^{+/Zα1α2}$ animals. Similar levels of *Zbp1* mRNA and ZBP1 protein were expressed at baseline and after IFN induction in cells expressing only wild-type ZBP1 ($Zbp1^{+/+}$), only mutant ZBP1 ($Zbp1^{Zα1α2/Zα1α2}$) or both ($Zbp1^{+/Zα1α2}$) (Figs 3A and EV3C). Consistent with our experiments based on ectopic ZBP1 expression (Figs 1 and 2), intact ZBDs of endogenous ZBP1 were required for cell death induced by MCMV-M45$^{mutRHIM}$ (Fig 3B and C). Cell death induced by TZ treatment was not affected by mutations in *Zbp1* (Figs 3B and EV3D). Furthermore, the levels of phosphorylated MLKL and MLKL oligomerisation were reduced in $Zbp1^{Zα1α2/Zα1α2}$ primary MEFs upon MCMV-M45$^{mutRHIM}$ infection (Figs 3D and EV3E), but not after TZ treatment (Fig 3E). Finally, virus growth and accumulation of the viral IE1 protein were enhanced in $Zbp1^{Zα1α2/Zα1α2}$ primary MEFs (Fig 3D and F). To test whether intact ZBDs are required to restrict virus replication *in vivo*, we infected $Zbp1^{Zα1α2/Zα1α2}$ knock-in mice with MCMV-M45$^{mutRHIM}$. After 5 days, we were able to recover infectious virus from the spleens of eight of 13 infected $Zbp1^{Zα1α2/Zα1α2}$ animals, while the spleens of all wild-type and heterozygous mice remained free of virus (Fig 3G). As expected, no differences in splenic virus titres were observed between the genotypes when mice were infected with wild-type MCMV (Fig 3G). These observations provide further evidence that recognition of nucleic acids, potentially in Z-conformation, by ZBP1 is required for the induction of necroptosis and virus restriction.

## MCMV-induced necroptosis requires viral RNA synthesis but not DNA replication

To define the molecular requirements of ZBP1 and necroptosis activation in MCMV-infected cells, we co-cultured NIH3T3 cells expressing wild-type and Zα1α2$^{mut}$ ZBP1 at a 2:1 ratio. The latter cells were labelled with a dye prior to co-culture. We infected co-cultures with MCMV-M45$^{wt}$ or MCMV-M45$^{mutRHIM}$ and, after 16 h, determined the ratio of wild-type ZBP1 and ZBP1-Zα1α2$^{mut}$ cells by flow cytometry (Fig 4A). Infection with MCMV-M45$^{wt}$ did not affect the ratio of these cells; however, MCMV-M45$^{mutRHIM}$ infection resulted in selective loss of wild-type ZBP1-expressing cells (Fig 4A). This observation shows that ZBP1-dependent cell death is cell autonomous and further indicates that infection generates an intracellular ZBP1 agonist, conceivably a nucleic acid in Z-conformation.

To determine whether viral DNA replication was required for ZBP1 activation, we treated infected cells with 50 μM of the

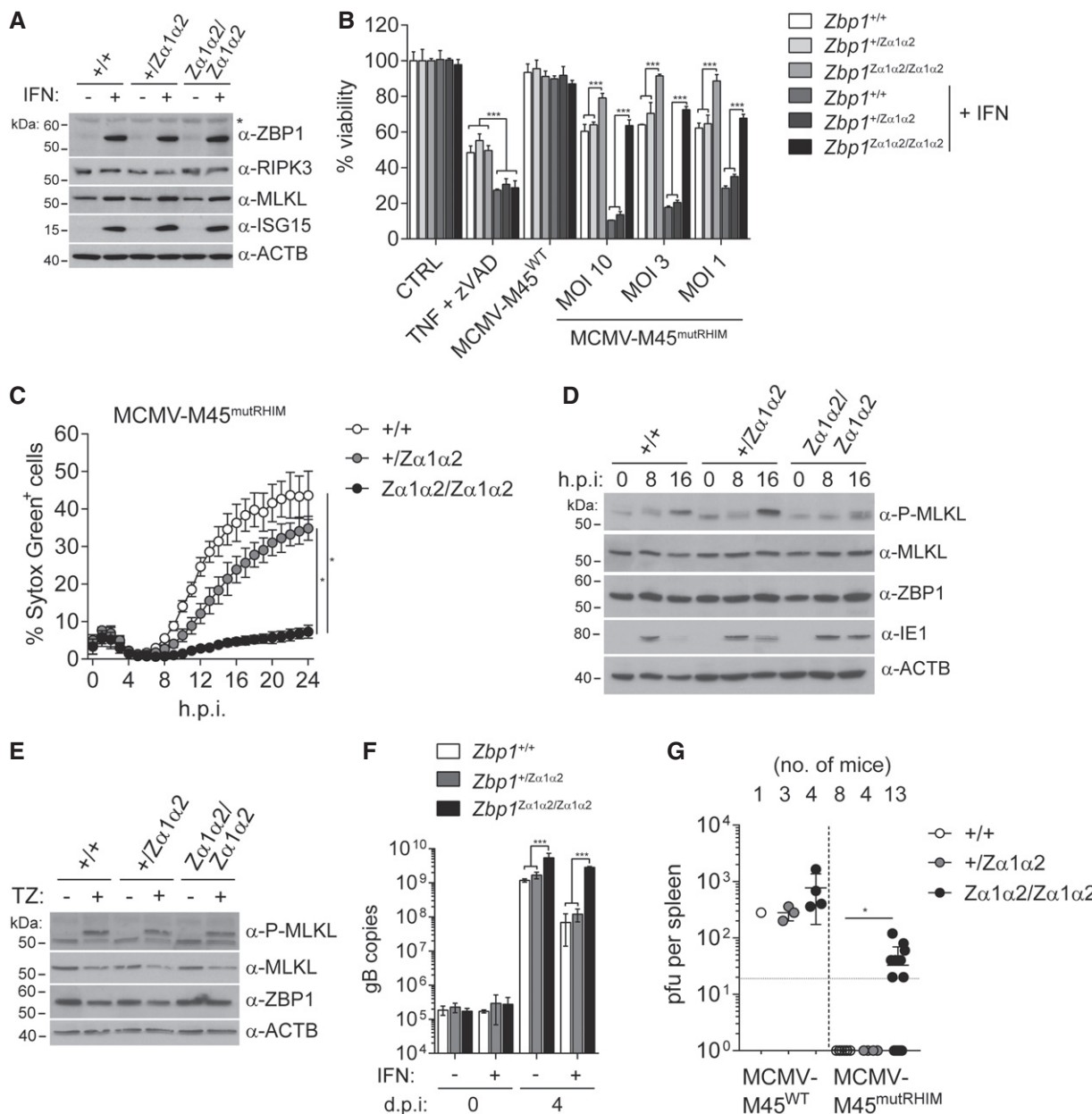

**Figure 3. Knock-in of ZBP1-Zα1α2^mut abrogates MCMV-triggered necroptosis.**

A    Primary MEFs of the indicated genotypes were treated or not with 100 U/ml of IFN-A/D for 16 h. Cell extracts were subjected to Western blot analysis using the indicated antibodies. Asterisk (*) indicates a non-specific band.

B    Primary MEFs of the indicated genotypes were treated or not with 100 U/ml of IFN-A/D for 16 h. Cells were treated with TZ or were infected with the indicated viruses (WT MCMV: MOI = 3). After 24 h, cell viability was assessed using CellTitre-Glo reagent. Values for untreated control cells were set to 100%.

C    Primary MEFs of the indicated genotypes. Cell death was monitored upon infection with MCMV-M45^mutRHIM (MOI = 3) using an in-incubator imaging platform (Incucyte) and the dye Sytox Green that stains dead cells.

D, E    Primary MEFs of the indicated genotypes were treated or not with 100 U/ml of IFN-A/D for 16 h. Cells were infected with MCMV-M45^mutRHIM at an MOI of 3 for 0, 8 and 16 h (D) or treated for 8 h with TZ (E), and cell lysates were analysed by Western blot.

F    Primary MEFs of the indicated genotypes were treated or not with 100 U/ml of IFN-A/D for 16 h. DNA samples were collected at 0 and 4 days after infection and analysed by qPCR. gB copy numbers were derived from a titration using defined amounts of gB plasmid.

G    Zα1α2-mutant Zbp1 knock-in (*Zbp1^Zα1α2/Zα1α2*), heterozygous (*Zbp1^+/Zα1α2*) and wild-type (*Zbp1^+/+*) littermate mice were infected with 2 × 10^6 pfu MCMV-M45^WT (one experiment) or MCMV-M45^mutRHIM (pooled data from two experiments using independent virus stocks) by intraperitoneal injection. Five days post-infection mice were sacrificed and viral titres in spleens were determined by plaque assay. Each dot represents one mouse. The dotted horizontal line depicts the detection limit.

Data information: If not indicated otherwise, data are representative of two or more independent experiments. Panels (B, C, F and G) show mean ± SD (*n* = 3 in B, C, F). *P < 0.05, ***P < 0.001; two-way ANOVA (panels B, C and F) and Mann–Whitney *t*-test (panel G). See also Fig EV3.

Source data are available online for this figure.

nucleotide analogue ganciclovir (Smee *et al*, 1995). Although this effectively prevented MCMV replication in NIH3T3 cells and immortalised MEFs (Fig EV4A), virus-induced cell death in wild-type ZBP1-reconstituted cells remained unaffected (Fig 4B). UV inactivation of MCMV-M45^mutRHIM abrogated viral replication and expression of viral genes (Figs 4C, and EV4A and B). At the same time, CXCL10 protein and ISG mRNA induction, which are dependent on STING (Figs 2G and EV2C), remained normal (Figs 4C and EV4B). Importantly, UV-treated MCMV did not induce cell death (Fig 4B). Together with the result using the DNA replication inhibitor ganciclovir, this suggested that viral RNA transcription was required for ZBP1-dependent necroptosis. Indeed, inhibition of *de novo* RNA synthesis using actinomycin D (ActD) prevented cell death induced by MCMV-M45^mutRHIM in NIH3T3 cells expressing wild-type ZBP1 (Fig 4D). Similarly, ActD but not the translation inhibitor cycloheximide (CHX) blocked MCMV-M45^mutRHIM induced cell death in primary MEFs (Fig 4E). We also interfered with expression of the viral immediate early protein IE3 that is required for expression of early and late genes (Angulo *et al*, 2000). siRNA-mediated depletion of IE3 was confirmed using MCMV-IE1/3-GFP infection (Martínez *et al*, 2010; Fig EV4C and D). IE3 depletion led to diminished cell death upon infection of NIH3T3 cells expressing wild-type ZBP1 with MCMV-M45^mutRHIM (Fig 4F). MCMV-M45^mutRHIM infection predisposes NIH3T3 cells to TNF-induced cell death independent of ZBP1 expression (Fig EV4E). This predisposition is likely due to expression of the MCMV protein M36, which blocks caspase-8 and in combination with TNF treatment triggers necroptosis (Guo *et al*, 2015; Omoto *et al*, 2015). IE3 depletion did not influence cell death triggered by MCMV-M45^mutRHIM and TNF, indicating that IE3 depletion did not change sensitisation to necroptosis (Fig EV4F). MCMV induces a potent IFN response (Fig 2E and F), and IFN-induced endogenous transcripts could contribute to ZBP1 activation. To test this, we cultured wild-type ZBP1-reconstituted MEFs in the presence of neutralising antibodies against the type I IFN receptor (anti-IFNAR1). Blockade of IFNAR1 signalling did not affect MCMV-M45^mutRHIM-induced cell death, while the expression of the ISGs *Ifit1* and *Ifi44*, but not of *Ifnb*, was reduced upon anti-IFNAR1 treatment (Fig EV4G and H). Taken together, these data show that MCMV

RNA transcripts generated in infected cells contribute to the activation of ZBP1 and necroptosis.

To further substantiate this conclusion, we employed the PAR-CLIP protocol (Fig 4G; Hafner *et al*, 2010). In brief, NIH3T3 cells expressing ZBP1-3xFLAG were pulsed with photo-activatable nucleoside analogues (4SU or 6SG) and were then infected with MCMV-M45^mutRHIM. Eight hours post-infection, cells were treated with UV-A light (365 nm), which only cross-links newly synthesised RNA that contains 4SU or 6SG to proteins. Cells were lysed and treated with RNase A. In this setting, small fragments of RNA are protected from degradation by cross-linking to proteins. ZBP1 was then immunoprecipitated using α-FLAG antibody and stringent conditions (see Materials and Methods). The RNA-ZBP1 complex was visualised by $^{32}$P labelling of the RNA, followed by gel electrophoresis, blotting and autoradiography (Fig 4G). Indeed, we observed bands corresponding to the molecular weight of ZBP1 (60 kDa) when MCMV-infected cells were treated with 4SU or 6SG (Fig 4G). To investigate whether cross-linking of newly synthesised RNA required intact ZBDs, NIH3T3 cells expressing wild-type or mutant ZBP1 were infected with MCMV-M45^WT in the presence of 6SG. Wild-type M45-expressing virus was used to prevent loss of wild-type ZBP1-expressing cells due to necroptosis. After cross-linking, we observed enhanced ZBP1-RNA complex formation with wild-type ZBP1 compared to mutant ZBP1, confirming that ZBDs are important for RNA binding (Fig 4H). These finding supports the notion that ZBP1 binds newly synthesised RNA in infected cells.

## Ectopic expression of ZBP1 predisposes cells to necroptosis

Surprisingly, we observed that expression of ZBP1 in NIH3T3 cells triggered spontaneous cell death upon caspase inhibition with zVAD (Figs 5A and EV5A). zVAD had a similar, albeit less pronounced, effect in immortalised MEFs (Fig 5A). Cell death in zVAD-treated cells expressing ZBP1 required intact ZBDs (Figs 5A and EV5A). Phosphorylation of MLKL at 16 h post-treatment confirmed that cells died by necroptosis (Fig EV5B). Furthermore, treatment with the RIPK3 inhibitor GSK'872 prevented cell death in zVAD-treated cells (Fig 5B). Viability was also restored by Nec-1 treatment,

**Figure 4. ZBP1 binds newly synthesised RNA in MCMV-infected cells.**

A   NIH3T3 cells expressing mutant ZBP1 were labelled with CellTrace Violet and were then co-cultured with cells expressing wild-type ZBP1 at a 1:2 ratio. Cells were then infected as indicated (MOI 1, 3 and 10) and analysed by flow cytometry after 16 h. Representative FACS histograms (MOI 3) are shown (left), and ratios of wild-type to mutant cell numbers were calculated (right).

B   Immortalised *Zbp1*^−/− MEFs were infected with MCMV-M45^mutRHIM in the presence or not of 50 µM ganciclovir (GCV) or with UV-treated virus at an MOI of 10. After 16 h, cell viability was assessed using CellTitre-Glo reagent. Values for untreated control cells were set to 100%.

C   Using supernatants and RNA samples from (B), CXCL10 protein levels and *Ifi44* and *M36* mRNA levels were determined by ELISA and RT–qPCR, respectively.

D   Cell death was monitored upon infection of NIH3T3 cells expressing wild-type ZBP1 with MCMV-M45^mutRHIM (MOI = 3) using an in-incubator imaging platform (Incucyte) and the dye YOYO-3 that stains dead cells. Actinomycin D (ActD) was added (5 µg/ml) at the indicated time points.

E   Primary MEFs of the indicated genotypes were pre-treated with 100 U/ml IFN-A/D for 16 h and were then treated with TZ, CHX (50 µg/ml) or ActD (5 µg/ml) in the presence or absence of MCMV-M45^mutRHIM. Cell viability was assessed as in (B).

F   NIH3T3 cells expressing wild-type ZBP1 were transfected with the indicated siRNAs and were then infected with MCMV-M45^mutRHIM. Cell viability was assessed as in (D).

G   Cells were treated with 100 µM 4SU or 6SG and infected with MCMV-M45^mutRHIM for 8 h. Binding of newly synthesised RNA to ZBP1 was analysed by PAR-CLIP (see Materials and Methods). Shown is an autoradiogram after electrophoresis and blotting of ZBP1 immunoprecipitates.

H   NIH3T3 cells expressing wild-type or mutant ZBP1 were treated with 100 µM 6SG and were infected with MCMV-M45^WT at an MOI of 3 for 8 h. Binding of ZBP1 to RNA was analysed as in (G).

Data information: Data are representative of three or more independent experiments. Panels (A–F) show mean ± SD (*n* = 3). *P < 0.05, **P < 0.01, ***P < 0.001; two-way ANOVA. See also Fig EV4.

Source data are available online for this figure.

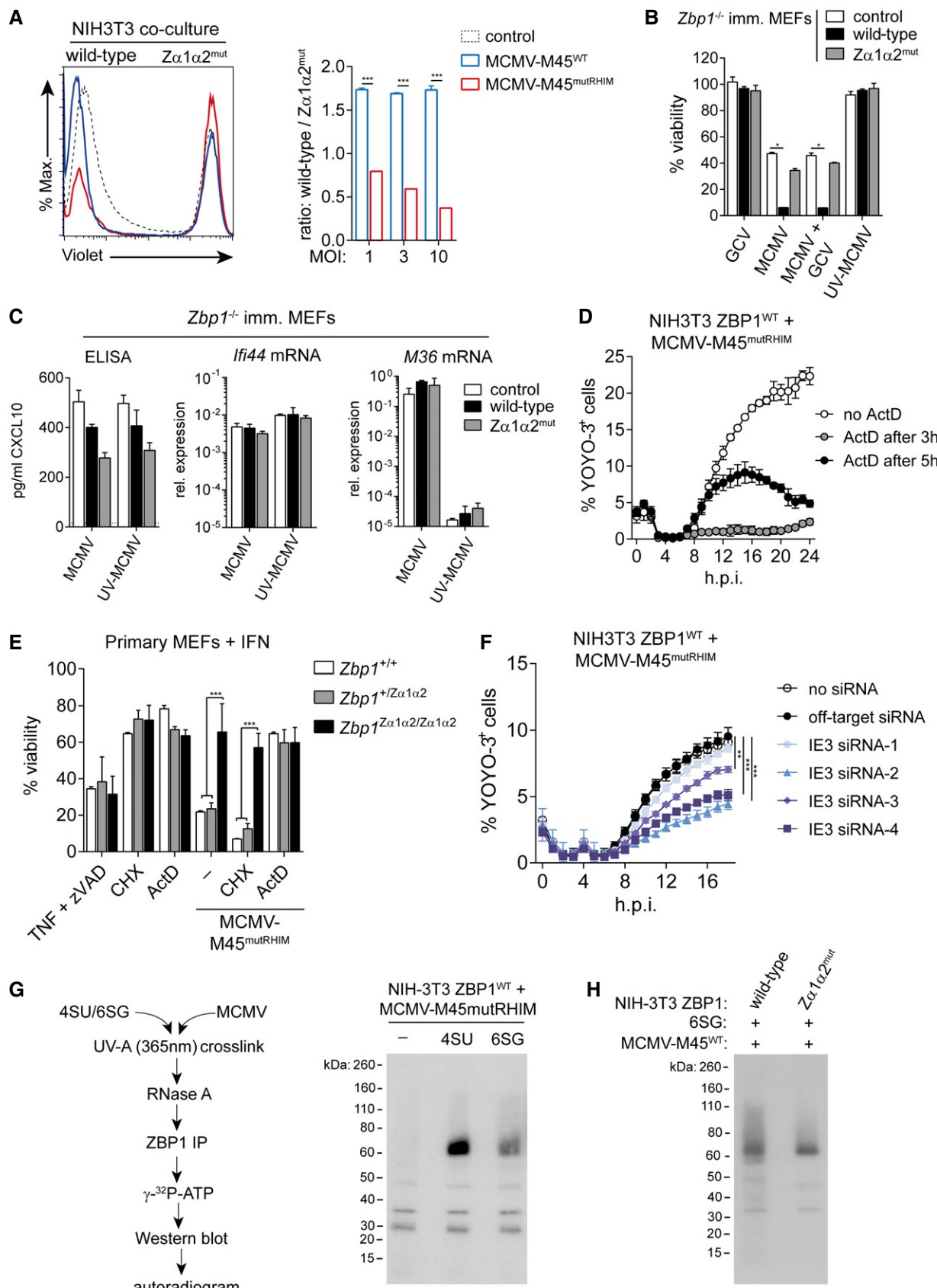

**Figure 4.**

suggesting that spontaneous ZBP1-dependent cell death induced by caspase inhibition additionally requires the activity of RIPK1 (Fig 5B).

Next, using a polyclonal cell population, we generated NIH3T3 clones expressing wild-type ZBP1 at different levels by limiting dilution (Figs 5C, and EV5C and D). Expression of ZBP1 at both low and high levels sensitised cells to killing by MCMV-M45$^{mutRHIM}$ (Fig 5D). In contrast, cell death triggered by ectopic ZBP1 and zVAD required high levels of expression (Fig 5D). To further explore this observation, we transduced cells expressing wild-type or Zα1α2$^{mut}$ ZBP1 with M36 that blocks caspase-8-mediated apoptosis. This led to selective loss of cells expressing wild-type ZBP1 and M36 (Fig 5E). Western blot analysis further indicated that expression of wild-type ZBP1 and M36 was mutually exclusive, indicating death of cells expressing these two proteins together (Fig 5F). Taken together, these observations suggested that cells contain endogenous nucleic acids that engage ZBP1 if present at sufficiently high levels. This then leads to necroptosis in conditions where caspase-8 activity is blocked.

### ZBP1 binds endogenous RNA

To further explore the observation that ZBP1 triggers necroptosis in response to endogenous nucleic acids, we used recombinant ZBP1 protein as a staining reagent in uninfected cells. We expressed 3xFLAG-tagged ZBP1 and ZBP1-Zα1α2$^{mut}$ in HEK293T cells and purified the proteins by immunoprecipitation (Fig EV6A). We then incubated uninfected, fixed and permeabilised HEK293T cells with purified ZBP1 protein, followed by staining with a secondary α-FLAG antibody. Cells were analysed by flow cytometry. Recombinant wild-type ZBP1 stained cells, whereas ZBP1-Zα1α2$^{mut}$ showed only background staining (Fig 6A, left panel). Interestingly, the labelling of cells with wild-type ZBP1 was sensitive to RNase but not to DNase treatment (Fig 6A, middle and right panel). These observations suggest that ZBP1, *via* its ZBDs, binds endogenous RNA, potentially in the Z-conformation.

Next, we used cross-link immunoprecipitation (CLIP, Fig 6B) to further explore the binding of endogenous nucleic acids to ZBP1 (Darnell, 2010). In brief, we cross-linked nucleic acids and proteins using UV-C light (254 nm) in uninfected NIH3T3 cells expressing ectopic ZBP1-3xFLAG. Cell extracts were then treated with DNase and/or RNase. This degrades free nucleic acids leaving intact fragments of DNA or RNA cross-linked to proteins. Next, we immunoprecipitated ZBP1 using α-FLAG antibodies (Fig EV6B). Nucleic acids present in the precipitate were then end-labelled using $^{32}$P, followed by Western blot and autoradiography (Fig 6B). Using high amounts of RNase A, we observed a major band in the autoradiogram corresponding in size to ZBP1 (sample 3 in Fig 6B). In samples without RNase A, this band was not visible; instead, a smear of higher molecular weight was apparent (sample 1). These patterns were not changed in samples containing DNase I (samples 2 and 4). These findings suggest that ZBP1 binds RNA and not DNA. At the high RNase A concentrations used here, a very short stretch of RNA is likely to be protected by ZBP1, explaining the sharp band in lane 3 in Fig 6B. In the absence of RNase, this band shifted to a slower migrating smear, indicating the presence of longer RNA fragments bound to ZBP1. Indeed, when we used a range of decreasing RNase A concentrations, we observed a concomitant increase in the amount of $^{32}$P-labelled material migrating more slowly (Figs 6C and EV6C). Importantly, this shift was markedly reduced when we used uninfected cells expressing ZBP1-Zα1α2$^{mut}$ despite equal efficiency of protein precipitation (Figs 6D and EV6D). Taken together, these data show that ZBP1 binds endogenous RNA but not DNA in uninfected cells.

## Discussion

Mammalian cells autonomously detect virus infection (Pichlmair & Reis e Sousa, 2007). Pattern recognition receptors (PRRs) sense virus invasion and recognise pathogen- or danger-associated molecular patterns (Medzhitov, 2009). Upon activation, these receptors induce immune responses, including the production of cytokines such as IFNs, chemokines and inflammatory mediators, as well as the triggering of programmed cell death (Pichlmair & Reis e Sousa, 2007; Mocarski *et al*, 2012, 2015; Lupfer *et al*, 2015). In many instances, nucleic acids such as viral RNA or DNA genomes trigger PRRs (Goubau *et al*, 2013). This raises an important question: How can PRRs distinguish between the RNAs and DNAs abundantly found in healthy, uninfected cells and nucleic acids introduced during infection? Several "solutions" to this problem have been described and include detection of nucleic acids in cellular compartments usually devoid of RNA or DNA, detection of covalent chemical modifications unique to viral RNAs and DNAs, and detection of specific sequences (Goubau *et al*, 2013; Roers *et al*, 2016).

Here, we propose that RNA in the unusual Z-conformation is a molecular pattern recognised by ZBP1. Indeed, we found that intact ZBDs were required for ZBP1-dependent induction of necroptosis in MCMV-infected cells. Several lines of evidence showed that ZBP1 recognises RNA, and not DNA, and that this RNA likely corresponds to viral transcripts newly synthesised after infection. Firstly, the induction of cell death could not be blocked by an inhibitor of the viral DNA polymerase (GCV). Secondly, UV-inactivated virus that failed to generate viral RNA transcripts also did not induce necroptosis. Thirdly, necroptosis induction was sensitive to ActD, which potently inhibits RNA transcription. Fourthly, knock-down of the viral IE3 gene—that drives expression of many other viral genes—reduced cell death. Lastly, PAR-CLIP analysis showed that ZBP1 cross-linked with newly synthesised RNA in infected cells.

Two recent studies published while this manuscript was in preparation suggest that ZBP1 senses influenza A virus (IAV) infection and induces cell death of IAV-infected cells (Kuriakose *et al*, 2016; Thapa *et al*, 2016). Our results confirm these two reports in a different infection model and resolve an important discordance: Thapa *et al* reported that ZBP1 senses IAV RNA (Thapa *et al*, 2016) while Kuriakose *et al* concluded that IAV proteins, particularly PB1 and NP, trigger ZBP1 (Kuriakose *et al*, 2016). Our results strongly suggest that ZBP1 is an RNA sensor. We used stringent cross-link and immunoprecipitation protocols and provide a detailed analysis of ZBP1 mutants unable to bind Z nucleic acids, including a knock-in model. It is conceivable that ZBP1 binds IAV PB1 and NP indirectly via viral RNA as both IAV proteins tightly associate with IAV RNA genomes and transcripts (Eisfeld *et al*, 2015). In the future, it will be interesting to infect ZBP1-Zα1α2$^{mut}$ knock-in mice with IAV to analyse the role of Z-RNA sensing and necroptosis in *in vivo* settings. Moreover, it will be worthwhile to explore other viral

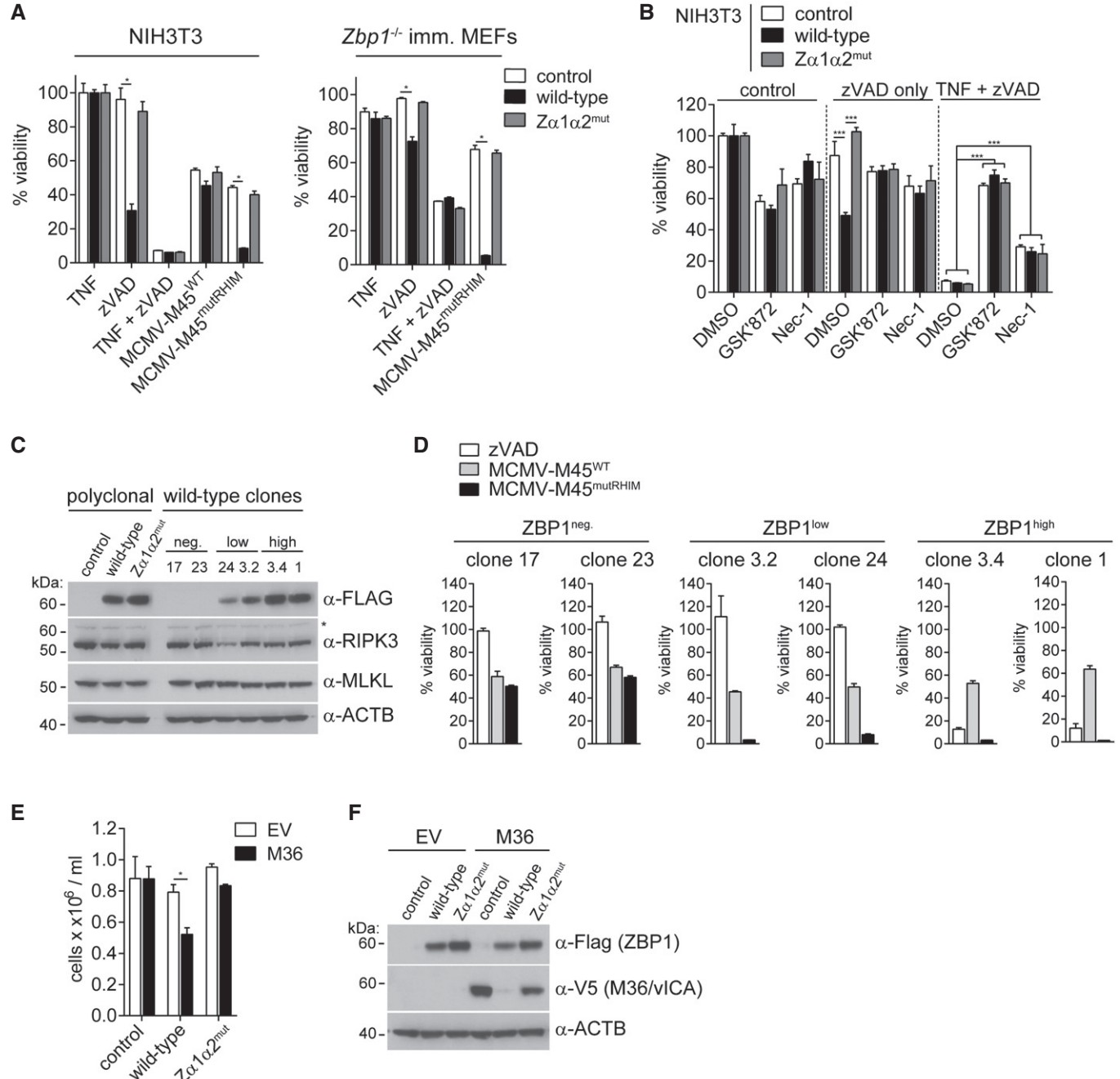

**Figure 5. High levels of ectopic ZBP1 expression sensitise cells to necroptosis.**

A   NIH3T3 cells or immortalised *Zbp1*$^{-/-}$ MEFs expressing wild-type or mutant ZBP1 were treated with 20 µM zVAD alone, 30 ng/ml TNF, or with TZ or were infected as indicated (MOI = 10). Cell viability was assessed as in Fig 1B after 16 h.

B   NIH3T3 cells reconstituted with wild-type or mutant ZBP1 were treated with 20 µM zVAD alone or TZ in combination with 3 µM GSK′872 (RIPK3 inhibitor) or 30 µM Nec-1 (RIPK1 inhibitor) or 0.1% DMSO as a control. Cell viability was monitored after 24 h as in Fig 1B.

C   Polyclonal and monoclonal NIH3T3 cell lines expressing no, low or high levels of ZBP1 were analysed by Western blot using the indicated antibodies. Asterisk (*) indicates a non-specific band.

D   Clonal cell lines from (C) were analysed as in (A).

E   *Zbp1*$^{-/-}$-immortalised MEFs expressing wild-type or mutant ZBP1 were transduced with empty or with M36 lentiviral vectors. Cell numbers were determined after 4 days.

F   Cells from (E) were tested by Western blot using the indicated antibodies.

Data information: Data are representative of two or more independent experiments. Panels (A, B, D and E) show mean ± SD (*n* = 3). *P < 0.05, ***P < 0.001; two-way ANOVA. See also Fig EV5.

Source data are available online for this figure.

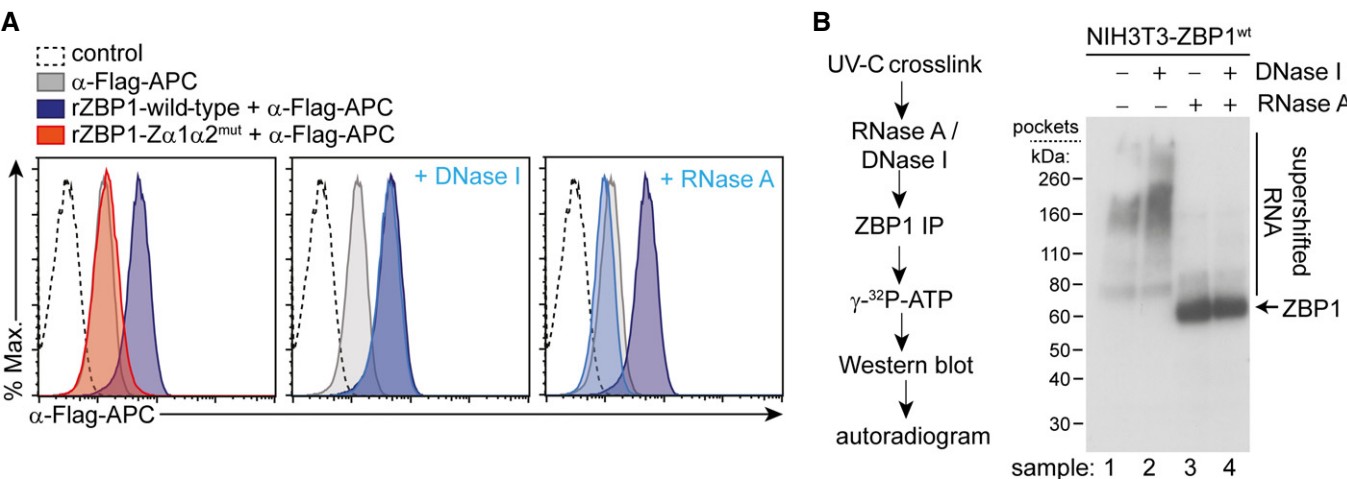

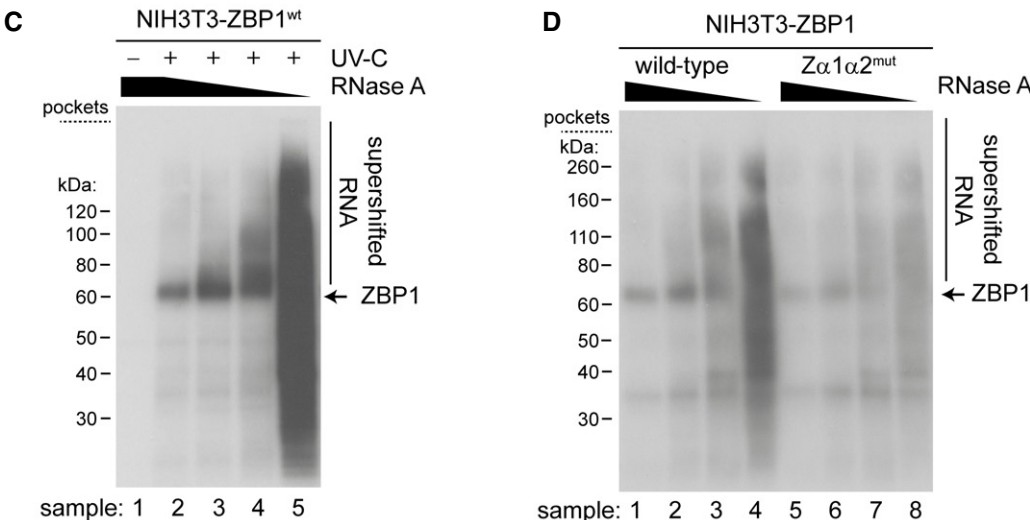

**Figure 6. ZBP1 binds endogenous RNA in uninfected cells.**

A   Fixed and permeabilised HEK293T cells were treated with DNase I or RNase A where indicated. Cells were then incubated with recombinant ZBP1-3xFLAG and were subsequently stained with α-FLAG-APC antibody and analysed by flow cytometry. Histograms representing the APC signal are shown.

B   An outline of the CLIP protocol is shown (left). NIH3T3 cells expressing wild-type ZBP1 were subjected to CLIP analysis using 20 U/ml DNase I and/or 20 U/ml RNase A. ZBP1 associated nucleic acids were labelled with $^{32}$P and resolved by gel electrophoresis. The autoradiogram is shown (right).

C   CLIP was performed as in (B) using different concentrations of RNase A. The autoradiogram shows supershifted RNA-ZBP1 complexes at intermediate RNase A concentrations.

D   CLIP analysis was done as in (B) and (C) using NIH3T3 cells expressing wild-type or mutant ZBP1.

Data information: For panels (B–D), sample numbers are indicated at the bottom of each lane. Data are representative of three or more independent experiments. See also Fig EV6.

Source data are available online for this figure.

infections. Indeed, necroptosis is also important during infection with human cytomegalovirus (HCMV) and herpes simplex viruses (HSV) 1 and 2 (Mocarski *et al*, 2015) and, in HIV-1 infection, necroptosis of infected T cells may contribute to pathogenesis (Gaiha *et al*, 2014; Pan *et al*, 2014).

Surprisingly, we found that cells expressing high levels of ZBP1 undergo necroptosis when caspase-8 is blocked and that ZBP1 cross-links to endogenous RNA in uninfected cells. This suggests a model in which ZBP1 continuously senses self-RNA, but only engages the necroptosis machinery when sufficient levels of

downstream signalling components such as RIPK3 become available, as is the case when caspase-8 is blocked. Indeed, loss of caspase-8 activity predisposes to RIPK3-dependent necroptosis (Kaiser *et al*, 2011; Oberst *et al*, 2011). Interestingly, ZBP1-dependent cell death in this setting required the activity of both RIPK1 and RIPK3. This is in contrast to MCMV-induced necroptosis, which occurs independently of RIPK1 (Upton *et al*, 2010). This may be important during development and in sterile inflammation. RIPK1-deficient mice display severe developmental defects and die around birth, and genetic ablation of RIPK1 in the skin leads to

inflammatory phenotypes (Silke *et al*, 2015). These observations have been linked to excessive RIPK3-MLKL-driven necroptosis (Silke *et al*, 2015). Two recent manuscripts showed that the defects of germline or keratinocyte-specific RIPK1-deficient mice are rescued by additional knock-out of ZBP1 (Lin *et al*, 2016; Newton *et al*, 2016). It is tempting to speculate that ZBP1 engages an endogenous Z-RNA in these settings and activates RIPK3 if inhibition by RIPK1 is lost. Our ZBP1-Z$\alpha$1$\alpha$2$^{mut}$ knock-in model will be an important tool to address this question.

It will also be interesting to extend our CLIP analysis by deep sequencing of ZBP1-associated nucleic acids. It is possible that ZBP1 binds GC-rich sequences given that GC-rich nucleic acids adopt the Z-conformation in the test tube under high salt conditions (Wang *et al*, 1979; Hall *et al*, 1984). It is noteworthy that in cells Z-RNA may not exist as a pre-formed structure but that this conformation is induced upon binding of ZBP1 and other ZBD-containing proteins to nucleic acids that are prone to adopt this structure, for example, due to their sequence content (Kim *et al*, 2011a). We would also like to point out that the RNA recognised by ZBP1 may correspond to both double-stranded and single-stranded molecules. The former may be the case in MCMV-infected cells, given that herpesviruses produce overlapping transcripts from both DNA strands (Rajcáni *et al*, 2004; Marcinowski *et al*, 2012; Juranic Lisnic *et al*, 2013). The latter may be the case when endogenous RNA is sensed; for example, ribosomal RNAs fold back and form base-paired hairpin structures that may be recognised by ZBDs (Feng *et al*, 2011).

Taken together, we demonstrate that binding of viral RNA to ZBP1 triggers necroptosis in infected cells. Whether these RNA molecules are indeed arranged in the Z-conformation requires further experimental testing. Additionally, endogenous nucleic acids may also adopt this unusual conformation and engage ZBP1. These data indicate that Z-RNA may be molecular pattern and that ZBP1 functions as its receptor, which induces inflammatory cell death.

# Materials and Methods

### Mice

All mice were on the C57Bl/6 background. *Tmem173*$^{-/-}$ (STING-deficient) mice were a gift from J Cambier (Jin *et al*, 2011). The *Zbp1* gene is located on chromosome 2 (GenBank NM_021394.2; Ensembl ENSMUSG00000027514). ZBP1-Z$\alpha$1$\alpha$2$^{mut}$ transgenic mice were generated by Cyagen (see Fig EV3A) and have been designated as Zbp1 < tm1.1Jreh> (MGI:5775152). See Appendix Supplementary Methods for further details. This work was performed in accordance with the UK Animals (Scientific Procedures) Act 1986 and institutional guidelines for animal care. This work was approved by a project license granted by the UK Home Office (PPL No. 40/3583) and was also approved by the Institutional Animal Ethics Committee Review Board at the University of Oxford.

### Cell culture

TLA HEK293T (Open Biosystems), NIH3T3 cells (kind gift from C. Reis e Sousa), and immortalised MEFs (generated in this study) were grown in DMEM (Sigma-Aldrich) supplemented with 10% heat-inactivated FCS and 2 mM L-Gln. Cells were grown at 37°C and

5% $CO_2$. Primary MEFs (kind gift from K. Ishii or generated in this study) were grown in DMEM 10% FCS, 2 mM L-Gln, 100 U/ml penicillin and 0.1 mg/ml streptomycin under low oxygen conditions (3% $O_2$) at 37°C and 5% $CO_2$. The absence of mycoplasma contamination was confirmed monthly for all cells.

### Plasmids and reagents

Mouse *Zbp1* was cloned from pCMV-SPORT6-Zbp1 (PlasmID clone: MmCD00314236, Harvard Medical School). A 3xFLAG-tag was added C-terminally to ZBP1 and TOPO-TA cloned into the pCR8/GW/TOPO entry vector (ThermoFisher). Z$\alpha$1 (N46A, Y50A) and Z$\alpha$2 (N122A, Y126A) mutant ZBP1 were cloned using overlap PCR. MCMV M36 was cloned from cDNA isolated from MCMV-infected NIH3T3 cells. ZBP1 and M36 were then Gateway cloned into pLenti6.3/TO/V5-DEST or pcDNA3.2/V5-DEST using the LR clonase II enzyme mix (ThermoFisher). The oligos used for cloning are listed in Appendix Table S1. The plasmids for human MDA5 (pcDNA3-3xFLAG-hMDA5), the IFN$\beta$ firefly luciferase reporter (p125Luc) and *Renilla* luciferase (Rluc) plasmids have been described before (Rehwinkel *et al*, 2010). The mouse RIPK3 expression plasmid (pCMV-SPORT6-mRIPK3) and the NF-$\kappa$B firefly luciferase reporter plasmid (pConLuc) were a gift from M. Bertrand and G. van Loo, respectively. The expression vector for mouse STING (pcDNA3-HA-mSTING) was a kind gift from R. Vance (Burdette *et al*, 2011). pGEMTeasy-MCMV-gB was a kind gift from E. Beuken. IFN-A/D was purchased from Sigma-Aldrich or R&D Systems. Recombinant mouse TNF was from PeproTech. zVAD (Z-VAD-FMK) was from Enzo Life Sciences. Actinomycin D was from Life Technologies and CHX was from Sigma-Aldrich. Nec-1 was from Enzo Life Sciences, and GSK'872 was from Calbiochem. Neutralising anti-IFNAR1 (clone MAR1-5A3) was from Biolegend.

### Virus production and infection

MCMV-M45$^{WT}$ (K181 strain) and MCMV-M45$^{mutRHIM}$ were a kind gift from E. Mocarski (Upton *et al*, 2010). MCMV was propagated in NIH3T3 cells or immortalised *Zbp1*$^{-/-}$ MEFs. See Appendix Supplementary Methods for further details. Lentivirus for ZBP1 and M36 reconstitution was produced in TLA HEK293T cells using pLenti6.3/TO/V5-DEST transducing vectors and the Virapower packaging mix (ThermoFisher).

### Cell viability assays

A total of 5,000 NIH3T3 cells or MEFs were seeded per well in 96-well plates. Twenty-four hours later, cells were infected with MCMV or treated with TNF and/or zVAD. Endpoint viability (generally 16 h post-infection/treatment) was determined by CellTiter-Glo (Promega) using a luminometer (GloMax, Promega). For kinetic analysis of cell death, an in-incubator imaging platform was used (IncuCyte ZOOM, Essen Bioscience; 10× objective, 1 scan per well in 96-well plates per hour; samples were imaged in triplicate wells). To measure viability, the cell-impermeant dyes YOYO-3 Iodide (0.25 μM) or Sytox Green (25 nM) (ThermoFisher) were added to the medium prior to imaging. At the end of the experiment, cells in control wells were lysed with 0.0625% Triton X-100 and total number of dye-positive cells were set to 100%. For NIH3T3

co-culture experiments, mutant ZBP1-expressing cells were labelled with 5 μM CellTrace Violet (Molecular Probes) for 20 min at 37°C. Cells were washed with medium and plated together with non-labelled wild-type ZBP1-expressing cells in 24 wells. Cells were then infected with MCMV. Sixteen hours post-infection, cells were harvested and the ratio of labelled vs. non-labelled cells was analysed by flow cytometry.

### Luciferase reporter assays and CXCL10 ELISA

A total of 150,000 TLA HEK293T cells were seeded in 24-well plates. Cells were transfected with NF-κB (50 ng) or IFNβ firefly (125 ng) and 25 ng of *Renilla* luciferase reporter constructs, and the indicated amounts of expression vectors using FuGENE 6 (Promega). Luciferase activity was measured 24 h after transfection using a luminometer (GloMax, Promega) and Dual-Glo reagents (Promega). CXCL10 ELISA was from eBioscience.

### qPCR and RT–qPCR

Cells were lysed in RLT buffer and homogenised using a Qiashredder (Qiagen), and RNA was purified using RNeasy columns with DNase I digestion (Qiagen). cDNA synthesis was performed with SuperScript II reverse transcriptase (Thermo Fisher) and oligo(dT) primers (Ambion). For DNA extraction, cells were lysed in AL-buffer and incubated with proteinase K at 56°C for 10 min, and DNA was purified using DNeasy blood and tissue columns (Qiagen). 15 ng DNA was amplified using TaqMan universal PCR (Applied Biosystems) or EXPRESS SYBR GreenER (ThermoFisher) master mixes on a 7,800 real-time PCR system (Applied Biosystems). mRNA expression data were normalised to GAPDH and analysed by the comparative Ct method. A plasmid carrying a fragment of the MCMV gB gene (pGEMTeasy-MCMV-gB) was used as a standard for determining the viral genome copy numbers (Vliegen *et al*, 2003). The lower detection limit was 30 molecules. The qPCR probes and primers used in this study are detailed in Appendix Table S2.

### Western blot

Cells were lysed in 25 mM Tris.HCl pH 7.4, 150 mM NaCl, 2 mM EDTA, 1% Igepal CA-630 (Sigma-Aldrich) and 5% glycerol, protease inhibitors (Cell Signalling) and phosphatase inhibitor cocktail 3 (Sigma). Protein lysates were cleared by centrifugation at 16,000 *g* for 10 min. Alternatively, cells were directly lysed in NuPAGE SDS–PAGE sample loading buffer (ThermoFisher). 50 mM of the reducing agent DTT was added to the lysates, except for non-reducing gels. Protein lysates were run on Bis-Tris SDS–PAGE gels (NuPAGE, ThermoFisher). Primary antibodies used for Western blotting are listed in Appendix Table S3. Anti-rabbit, anti-rat or anti-mouse secondary antibodies coupled to HRP were from GE Healthcare.

### Flow cytometry and fluorescence microscopy

For intracellular ZBP1 staining, cells were fixed and permeabilised in Cytofix/Cytoperm (BD biosciences) according to manufacturer's instructions. When recombinant ZBP1 was used (Fig 6A), cells were incubated with 0.3 μg ZBP1-3xFLAG for 1 h in Perm/Wash buffer. In some instances, cells were pre-incubated for 30 min at 37°C with

30 U Purelink RNase A (Invitrogen) or Turbo DNase I (Ambion). Anti-FLAG-APC (Abcam, clone M2) was used to detect ZBP1-FLAG. Samples were analysed in FACS buffer (PBS, 2 mM EDTA, 1% FCS). 1 μg/ml DAPI (Sigma-Aldrich) or Live/Dead Fixable Violet (Molecular Probes) was used to exclude dead cells. Data were acquired on Beckmann Coulter CyAn or BD Biosciences LSRFortessa flow cytometers. FCS files were analysed using FlowJo V.10.0.8. For fluorescence microscopy, polyclonal ZBP1-expressing NIH3T3 cells were seeded on poly-ᴅ-lysine (Sigma-Aldrich)-coated coverslips (Lab-Tek). Cells were fixed using 4% paraformaldehyde (Thermo-Fisher) and blocked and permeabilised for 1 h in PBS + 5% BSA (Sigma-Aldrich) + 0.1% saponin (Sigma-Aldrich). Primary antibody (anti-FLAG-HRP, clone M2, Sigma-Aldrich) and detection antibodies (anti-HRP-Alexa Fluor 594, Jackson Immunoresearch) were incubated for 1 h in PBS + 5% BSA + 0.1% saponin (Sigma-Aldrich). Images were acquired using a Zeiss Axiovert S100 inverted fluorescence microscope.

### IE3 knockdown

siRNAs targeting MCMV IE3 were directed against exon 5 of the IE1/3 gene and were designed using siDESIGN Center from Dharmacon (GE Healthcare). siRNAs were ordered from Dharmacon, and the sequences are detailed in Appendix Table S4. A total of 5,000 NIH3T3 cells were seeded per well in 96-well plates. 25 nM siRNAs were transfected using DharmaFECT 1 (Dharmacon, GE Healthcare) according to manufacturer's instructions. One hour later, transfection mix was washed away, and cells were infected with MCMV.

### (PAR)-CLIP

PAR-CLIP and CLIP protocols were adapted from (Hafner *et al*, 2010; Huppertz *et al*, 2014). Further details are provided in Appendix Supplementary Methods.

### ZBP1 purification

Wild-type and Zα1α2ᵐᵘᵗ mouse ZBP1 were purified based on the protocol published in Sun *et al* (2013). See Appendix Supplementary Methods for further details.

### Statistics

Statistical analysis was performed in GraphPad Prism v7.00 as detailed in the figure legends.

**Expanded View** for this article is available online.

## Acknowledgements

The authors thank K. Ishii for *Zbp1*⁻/⁻ MEFs, E. Mocarski for MCMV-M45ᵐᵘᵗᴿᴴᴵᴹ virus, A. Pichlmair for the α-ZBP1 antibody and Q. Tang for MCMV-IE1/3-GFP and the α-IE1 antibody. The authors further thank A. Jackson and members of the Rehwinkel laboratory for critical discussions and reading of the manuscript draft. This work was funded by the UK Medical Research Council (MRC core funding of the MRC Human Immunology Unit) and by the Wellcome Trust (grant number 100954). KBR and JWU were funded by the Cancer Prevention and Research Institute of Texas (CPRIT), R1202. JM was a recipient of an EMBO long-term postdoctoral fellowship and was also

supported by Marie Curie Actions (EMBOCOFUND2010, GA-2010-267146). *Tmem173*$^{-/-}$ mice were provided by J. Cambier and are subject to materials transfer agreements.

## Author contributions

JM and JR conceived the study, designed experiments, analysed data and wrote the manuscript. JM performed all experiments. LL and AB provided technical help and analysed data. KBR and JWU provided MCMV-M45$^{mutRHIM}$ virus for the experiment shown in Fig 3G. All authors read and approved the final manuscript.

## Conflict of interest

The authors declare that they have no conflict of interest.

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
