## [Review Process File · The EMBO Journal]

Manuscript EMBO-2017-96476

Sensing of viral and endogenous RNA by ZBP1/DAI induces necroptosis

Jonathan Maelfait, Layal Liverpool, Anne Bridgeman, Katherine B Ragan, Jason W Upton, Jan Rehwinkel

Corresponding author: Jonathan Maelfait & Prof. Jan Rehwinkel, University of Oxford

Review timeline:	Submission date:	07 January 2017
	Editorial Decision:	15 February 2017
	Revision received:	15 May 2017
	Editorial Decision:	24 May 2017
	Revision received:	04 June 2017
	Accepted:	13 June 2017

Editor: Karin Dumstrei

Transaction Report:

1st Editorial Decision

15 February 2017

Thank you for submitting your manuscript to The EMBO Journal. I am sorry for the slight delay in getting back to you, but I have now received the full set of comments on your manuscript.

As you can see below the referees appreciate the analysis. While referee #3 is not convinced that the advance provided over previous work is sufficient to consider publication here, referees # 1 and 2 are more supportive. I see the points raised by referee #3, but I am also in agreement with referees #1 and 2 that the analysis extends previous work. I would therefore like to invite you to submit a suitable revised manuscript. You would need to address the specific points raised, but not the further reaching point brought up by referee #3 (to look at ZBP1 in RIPK1-deficient mice).

Let me know if we need to discuss anything further.

When preparing your letter of response to the referees' comments, please bear in mind that this will form part of the Review Process File, and will therefore be available online to the community. For more details on our Transparent Editorial Process, please visit our website: http://emboj.embopress.org/about#Transparent_Process

I thank you for the opportunity to consider your work for publication. I look forward to your revision.

REFEREE REPORTS

Referee #1:

Maelfait et al. describe that recognition of viral and endogenous Z-RNA by ZBP1 is responsible for induction of necroptosis. By employing the mutant ZBP1 that supposedly abrogates binding to Z-DNA/RNA, the authors found that ZBP1 indeed requires its association to trigger cell death. Intriguingly, they revealed that RNA synthesis, not DNA replication or translation, is necessary for MCMV-induced necroptosis. In addition, the authors identified that ZBP1 recognizes endogenous RNA, which appears to be required for death of uninfected cells with high expression of ZBP1. Collectively, this research highlights the importance of ZBP1 sensing both endogenous and viral RNA, rather than DNA, to regulate cell death. In summary, the study by Maelfait et al. demonstrates the role of RNA-binding properties of ZBP1 in its necroptosis-inducing function in response to viral infection. This is very good study and, overall, the conclusions are well supported by the presented data. Yet some questions need to be addressed to fully substantiate their conclusions. I therefore have the following comments that need to be addressed to further support and strengthen the conclusions.

Major comments:

1. The experiment presented in Figure 4G shows that ZBP1 binds to newly synthesised RNA upon infection, however, it does not prove that this is dependent on the $Z\alpha1Z\alpha2$ domains, and this needs to be evaluated.
2. The authors need to explain how it is possible that necroptosis in uninfected cells does not occur despite the binding of ZBP1 to endogenous Z-RNA whilst infected cells die by viral RNA-bound ZBP1.
3. The authors claimed that "MCMV-induced necroptosis requires viral RNA synthesis but not DNA replication". However, it is reasonable to assume that some endogenous RNA synthesis should be upregulated upon viral infection. For instance, IFN-responsive gene transcripts are induced by autocrine IFN signaling. Can the authors exclude the involvement of endogenous RNA?
4. The authors did not experimentally prove that cell death induced by MCMV and ectopic ZBP1 expression is indeed necroptotic, i.e. cell death mediated by RIPK3 and MLKL. This analysis needs to be provided.

Minor comment:

Figure 1H: The line graph representing the $Z\alpha1$ mutant is barely visible. Please change the grey shading to color/different symbols instead.

Referee #2:

In this work Rehwinkel have explored the role and mechanism of action of ZBP1 in necroptosis. The work follows a previous paper demonstrating that ZBP1 induces necroptosis during infection with the MCMV M45mutRHIM mutant (Upton, 2012). The authors confirm the previous findings, and extend the current knowledge, by showing that ZBP1 requires a functional Z-RNA/DNA-binding domain to induce necroptosis, thus formally demonstrating that ZBP1 is a bona-fide innate nucleic acid-sensing receptor. The work is well designed, the results are solid, and the conclusions are generally supported by the data. I find this work highly interesting. The only major point from my side is that the authors should characterize the molecular nature of the ZBP1 agonist (cellular or viral) in more details, if they want to put it in the title.

MAJOR POINTS

1. Although the authors put "Z-RNA" in the title, and claim to have demonstrated this molecular signature to be a necroptosis-inducing ZBP1 agonist, they do not formally demonstrate this. Data should be provided from experiments where wt and $Zbp1^{-/-}$ cells are treated with synthetic nucleic acid species, including Z-RNA and Z-DNA, and viability is evaluated. In the absence of such data, the authors cannot draw the conclusion that Z-RNA is sensed by ZBP1 to induce necroptosis.

2. Along the same lines, the title mentions "viral Z-RNA". There are no data on viral RNA binding to ZBP1. It is surprising that all the binding experiments are from non-infected cells, and somewhat confusing why the authors make this switch in experimental set-up for the binding studies.
3. There is little data linking human ZBP1 to Z-RNA-induced necroptosis in human cells. This is important, since the nucleic acid field has revealed significant specie differences.
4. The authors have generated a mouse strain harboring a form of ZBP1 unable to bind Z-RNA/DNA. The work would gain significantly, if this strain was used to generate in vivo data to back up the in vitro experiments.

Referee #3:

In this manuscript Maelfait and colleagues study the role of ZBP1 for the detection of MCMV. In brief, their most important findings can be summarized as follows: Using an MCMV mutant virus that does not prevent the induction of necroptosis, the authors show that ZBP1 triggers necroptosis following MCMV infection. This phenomenon requires intact N-terminal nucleic acid binding domains, as mutants of ZBP1 in that region show no such activity. Furthermore, using various independent approaches, the authors can show that ZBP1 detects de novo transcribed viral RNA, but not DNA.

Upton and colleagues (Cell Host & Microbe 2012) already provided a comprehensive analysis of the role of ZBP1 in the context of MCMV infection (using the same mutant MCMV that is defective in blocking ZBP1). In their study it was shown that ZBP1 was critically involved in inducing necroptosis upon MCMV infection. Moreover, in vivo studies revealed that ZBP1-dependent virus detection played an important role in containing viral replication. More recently, Thapa and colleagues (Cell Host & Microbe 2016) convincingly showed that ZBP1 was also important for the recognition of Influenza A Virus leading to necroptosis induction in vitro and thereby containment of viral replication in vivo. In this study, it was furthermore demonstrated that ZBP1 directly bound to viral RNA. These studies thereby established ZBP1 as a bona fide PRR that binds RNA. This current manuscript now extends these findings to MCMV, by showing that MCMV-derived RNA transcripts are required to trigger ZBP1 activation leading to necroptosis.

An interesting observation that the authors make is that ZBP1 binds to endogenous RNA under conditions where it is overexpressed (Fig. 6). The authors suggest that endogenous RNA binding by ZBP1 might be involved in triggering necroptosis once caspases are inhibited (see Fig. 5), yet these studies were performed in cells expressing non-physiological amounts of ZBP1, whereas WT cells (with normal levels of ZBP1) don't show this phenotype. As such, the physiological relevance of these findings remains unclear.

Altogether, the experiments are well performed and well described. However, in light of previously published work on this topic, this manuscript provides only little, incremental insight into the biology of ZBP1. As such, I do not find this manuscript suitable for publication at The EMBO Journal in its current form. I would suggest that the authors use their mouse model (ZBP1-*Zα1α2mut*) to study the role of ZBP1 in the context of perinatal lethality of RIPK1-deficient mice (see recent work by the labs of Dixit and Pasparakis). This way they could address whether ZBP1 detects endogenous RNAs under physiological conditions (which is counteracted by RIPK1 under steady state conditions).

Minor point:

Without direct proof, I wouldn't state that ZBP1 detects Z-form RNA (at present, the authors have only shown that ZBP1 binds RNA, yet whether the bound RNA is of Z-form has not been shown).

1st Revision - authors' response

15 May 2017

Response to referee #1:

Maelfait et al. describe that recognition of viral and endogenous Z-RNA by ZBP1 is responsible for induction of necroptosis. By employing the mutant ZBP1 that supposedly abrogates binding to Z-

DNA/RNA, the authors found that ZBP1 indeed requires its association to trigger cell death. Intriguingly, they revealed that RNA synthesis, not DNA replication or translation, is necessary for MCMV-induced necroptosis. In addition, the authors identified that ZBP1 recognizes endogenous RNA, which appears to be required for death of uninfected cells with high expression of ZBP1. Collectively, this research highlights the importance of ZBP1 sensing both endogenous and viral RNA, rather than DNA, to regulate cell death.

In summary, the study by Maelfait et al. demonstrates the role of RNA-binding properties of ZBP1 in its necroptosis-inducing function in response to viral infection. This is very good study and, overall, the conclusions are well supported by the presented data. Yet some questions needs to be addressed to fully substantiate their conclusions. I therefore have the following comments that need to be addressed to further support and strengthen the conclusions.

We thank the reviewer for his / her positive and constructive assessment of our manuscript.

Major comments:

1. The experiment presented in Figure 4G shows that ZBP1 binds to newly synthesised RNA upon infection, however, it does not prove that this is dependent on the Za1Za2 domains, and this needs to be evaluated.

We agree with the reviewer that this is an important point that needed to be addressed. To evaluate whether intact Za domains of ZBP1 are required for binding to newly synthesized RNA, we have repeated the PAR-CLIP experiment shown in Figure 4G using NIH3T3 cells expressing wild type or Za1a2-mutant ZBP1 and the photoactivatable ribonucleotide 6SG. Initial experiments using MCMV-M45^{mutRHIM} were hampered by the selective loss of input material of cells transduced with wild type ZBP1 due to necroptosis. Although M45 blocks ZBP1-RIPK3 interactions, it is not predicted to prevent upstream binding of RNA to ZBP1. Therefore, to quantitatively compare ZBP1-RNA complex formation in both cell lines, we infected cells with MCMV-M45^{WT}, which blocks necroptosis, followed by PAR-CLIP analysis. As expected, the experiment shown in the new Fig 4H indicates enhanced crosslinking of RNA to wild type ZBP1.

2. The authors need to explain how it is possible that necroptosis in uninfected cells does not occur despite the binding of ZBP1 to endogenous Z-RNA whilst infected cells die by viral RNA-bound ZBP1.

We apologise to the reviewer if this aspect of the manuscript was not clearly explained. Necroptotic cell death is induced under conditions when caspase-8 activity is compromised (Kaiser et al, 2011; Oberst et al, 2011). During MCMV infection the viral M36 protein inhibits caspase-8 activation (and subsequent apoptosis) and concomitantly predisposes cells to necroptosis (Daley-Bauer et al, 2017). We show that during infection with MCMV-M45^{mutRHIM}, which does not block RIPK3-mediated necroptosis, viral RNA transcripts trigger ZBP1 mediated necroptosis. In uninfected cells we observed that ZBP1 associates with endogenous RNA. This interaction is, however, not sufficient to drive cell death as a block on caspase-8 activity is required to trigger necroptosis, for example by incubating cells with zVAD (Fig 5A) or transducing cells with M36 (Fig 5E and F). To make this point clearer, we have re-worded the discussion on page 17: "This suggests a model in which ZBP1 continuously senses self-RNA, but only engages the necroptosis machinery when sufficient levels of downstream signalling components such as RIPK3 become available, as is the case when caspase-8 is blocked."

3. The authors claimed that "MCMV-induced necroptosis requires viral RNA synthesis but not DNA replication". However, it is reasonable to assume that some endogenous RNA synthesis should be upregulated upon viral infection. For instance, IFN-responsive gene transcripts are induced by autocrine IFN signaling. Can the authors exclude the involvement of endogenous RNA?

We thank the reviewer for this question. To exclude a role for IFN induced transcripts, we have infected MEFs reconstituted with wild type ZBP1 with MCMV-M45^{mutRHIM} in the presence of type I interferon receptor (IFNAR) blocking antibodies. As expected, anti-IFNAR1 treatment reduced the expression of the ISGs *Ifi1* and *Ifi44*, and *Ifnb* transcript levels were not affected (*Ifnb* is not an ISG). Importantly, we did not observe a reduction in cell death after blocking type I IFN signalling.

These data suggest that IFN-induced transcripts do not contribute to ZBP1 induced cell death and have now been incorporated into the new manuscript in Fig EV4G and H.

4. The authors did not experimentally prove that cell death induced by MCMV and ectopic ZBP1 expression is indeed necroptotic, i.e. cell death mediated by RIPK3 and MLKL. This analysis needs to be provided.

As part of the initial submission, we had already shown phosphorylation and oligomerisation of MLKL, which are markers for necroptosis (Wallach et al, 2016), upon MCMV-M45^{mutRHIM} infection of NIH3T3 cells transduced with ZBP1 (Fig 2C and EV2B) or in primary MEFs (Fig 3D and EV3E). Similarly, we demonstrated phosphorylation of MLKL after treatment with zVAD-only in NIH3T3 cells ectopically expressing ZBP1 (Fig EV5B). To substantiate the evidence for necroptosis as the type of cell death, we have now extended our analysis by using chemical inhibitors of RIPK3 (GSK'872) and RIPK1 (Nec-1) activation. Inhibition of RIPK3 fully restored viability after virus infection and zVAD-only treatment. We have included these data in the new manuscript in Fig 2D (virus infection) and Fig 5B (zVAD-only). As reported previously (Upton et al, 2010), inhibition of RIPK1 did not rescue virus induced cell death (Fig 2D). Interestingly, cell death induced by zVAD-only was inhibited by Nec-1 treatment, indicating the RIPK1 activity is required in this setting. This may be important for the role of ZBP1 in sterile inflammation and is briefly discussed on page 17. Taken together, the biochemical data and the use of inhibitors strongly indicate that the type of cell death after virus infection and after caspase-8 inhibition is indeed necroptosis.

Minor comment:

Figure 1H: The line graph representing the Za1 mutant is barely visible. Please change the grey shading to color/different symbols instead.

We thank the reviewer for pointing this out. We have adjusted Fig 1H accordingly.

Response to referee #2:

In this work Rehwinkel have explored the role and mechanism of action of ZBP1 in necroptosis. The work follows a previous paper demonstrating that ZBP1 induces necroptosis during infection with the MCMV M45mutRHIM mutant (Upton, 2012). The authors confirm the previous findings, and extend the current knowledge, by showing that ZBP1 requires a functional Z-RNA/DNA-binding domain to induce necroptosis, thus formally demonstrating that ZBP1 is a bona-fide innate nucleic acid-sensing receptor. The work is well designed, the results are solid, and the conclusions are generally supported by the data. I find this work highly interesting. The only major point from my side is that the authors should characterize the molecular nature of the ZBP1 agonist (cellular or viral) in more details, if they want to put it in the title.

We thank the reviewer for his / her positive evaluation of our work. We are delighted to hear that the reviewer finds our study highly interesting and is convinced by its conceptual advance, namely that ZBP1 is a bona-fide sensor for RNA. We further agree with the reviewer's comment (see also major points 1 and 2 below) that the exact molecular nature of the ZBP1 ligand remains to be determined. Addressing this will require a lot of additional and time-consuming experimentation, such as deep sequencing and characterization of ZBP1-associated RNA. We believe that such studies are beyond the scope of this manuscript. To avoid over-stating our conclusions, we have changed the title of our manuscript to: Sensing of viral and endogenous RNA by ZBP1/DAI induces necroptosis. Throughout the text we have also removed statements that directly claim that ZBP1 senses Z-RNA. We have highlighted these changes in red.

Major points:

1. Although the authors put "Z-RNA" in the title, and claim to have demonstrated this molecular signature to be a necroptosis-inducing ZBP1 agonist, they do not formally demonstrate this. Data should be provided from experiments where wt and Zbp1^{-/-} cells are treated with synthetic nucleic acid species, including Z-RNA and Z-DNA, and viability is evaluated. In the absence of such data, the authors cannot draw the conclusion that Z-RNA is sensed by ZBP1 to induce necroptosis.

We thank the reviewer for this suggestion. Unfortunately, however, the proposed experiment is not feasible due to the inherent instability of the Z-RNA conformation (Hall et al, 1984). Indeed, under normal conditions, the A conformation of dsRNA is thermodynamically favoured over the Z conformation, which can only be stabilised by chemical modifications, increased temperature or non-physiological, high-salt buffer (Hardin et al, 1987; Popenda et al, 2004; Ross et al, 1989). For these reasons, stimulation of cells with synthetic Z-RNA is technically challenging if not impossible. Nonetheless, we have attempted transfections with the self-complementary RNA hexamer (CG)₃, which is prone to form Z-RNA under high salt concentrations in the test tube (Popenda et al, 2004). However, we did not observe ZBP1-dependent cell death. The conditions that stimulate Z-RNA formation in living cells are unknown. It is possible that RNA-protein interactions result in Z-RNA formation. For example, the ZBDs of ZBP1 may induce Z-RNA formation upon binding to A-form RNA sequences prone to form Z-RNA. We explicitly mention this point in the discussion (page 17-18). To avoid overstating our conclusions, we have – as discussed above – changed the title of the manuscript and made the necessary changes in the text (highlighted in red).

2. Along the same lines, the title mentions "viral Z-RNA". There are no data on viral RNA binding to ZBP1. It is surprising that all the binding experiments are from non-infected cells, and somewhat confusing why the authors make this switch in experimental set-up for the binding studies.

Several lines of evidence support the notion that newly synthesized RNA of viral origin stimulates ZBP1 upon MCMV infection. (i) Inhibition of transcription with actinomycin D at 3 or 5 hours post-infection prevents ZBP1 induced cell death (Fig 4D and E), and ZBP1 crosslinks to RNA in PAR-CLIP experiments (Fig 4G and H). These results show that ZBP1 binds to newly synthesised RNA in infected cells. (ii) Knockdown of IE3 (the viral trans-activator of early and late viral gene expression) suppresses cell death during infection (Fig 4F) but not cell death triggered by TNF (Fig EV4F), suggesting that this newly synthesised RNA is of viral origin. (iii) Supporting this notion is the observation that UV-inactivated virus, which does not generate viral transcripts, did not induce cell death (Fig 4B). (iv) New data in Fig EV4G and H show that blockade of host ISG transcription did not prevent virus-triggered cell death (see also reviewer 1 point 3). We therefore believe that the conclusion that viral RNAs contribute to activation of ZBP1 is justified.

3. There is little data linking human ZBP1 to Z-RNA-induced necroptosis in human cells. This is important, since the nucleic acid field has revealed significant specie differences.

We agree with the reviewer that this is an important question. We have worked hard to generate the required reagents such as hZBP1 constructs and hZBP1 knockout or reconstituted human cell lines (none of which were available in the lab); however, due to time constraints of the 90-day revision period, we have been unable to obtain these data. We apologise to the reviewer that we are not able to present human data at this point.

4. The authors have generated a mouse strain harboring a form of ZBP1 unable to bind Z-RNA/DNA. The work would gain significantly, if this strain was used to generate in vivo data to back up the in vitro experiments.

This is an important question. We have performed an initial experiment presented in the figure below. We infected $Z\alpha1\alpha2$ -mutant *Zbp1* knock-in mice, as well as heterozygous and wild type littermates, with MCMV-M45^{WT} and MCMV-M45^{mutRHIM}. Viral titres in the spleen were determined five days post-infection. In accordance with previously published data, wild type MCMV-M45^{WT} replication is equally efficient in all genotypes and MCMV-M45^{mutRHIM} did not infect mice with an intact *Zbp1* allele (Upton et al, 2012). However, M45-mutant virus was detected in spleens of 4 out of 8 *Zbp1*^{Z α 1 α 2/Z α 1 α 2} animals. These results indicate that RNA sensing by ZBP1 is required for MCMV restriction *in vivo*. Due to time constraints we were not able to breed our colony to sufficient numbers to generate more repeats of this experiment. Therefore, we have decided to not include these data in the revised manuscript.

ZDBs of ZBP1 are required for MCMV restriction *in vivo*. *Zα1α2*-mutant *Zbp1* knock-in mice (*Zbp1*^{Zα1α2/Zα1α2}), heterozygous (*Zbp1*^{+/Zα1α2}) and wild type (*Zbp1*^{+/+}) littermates were infected with 2×10^6 pfu MCMV-M45^{WT} or MCMV-M45^{mutRHIM} by i.p. injection. 5 days post-infection mice were sacrificed and viral titres in spleens were determined by plaque assay. The horizontal dotted line depicts the detection limit of 39 pfu. Each dot represents one mouse. The horizontal line and error bars represent mean \pm SD, respectively.

Response to referee #3:

In this manuscript Maelfait and colleagues study the role of ZBP1 for the detection of MCMV. In brief, their most important findings can be summarized as follows: Using an MCMV mutant virus that does not prevent the induction of necroptosis, the authors show that ZBP1 triggers necroptosis following MCMV infection. This phenomenon requires intact N-terminal nucleic acid binding domains, as mutants of ZBP1 in that region show no such activity. Furthermore, using various independent approaches, the authors can show that ZBP1 detects de novo transcribed viral RNA, but not DNA.

Upton and colleagues (Cell Host & Microbe 2012) already provided a comprehensive analysis of the role of ZBP1 in the context of MCMV infection (using the same mutant MCMV that is defective in blocking ZBP1). In their study it was shown that ZBP1 was critically involved in inducing necroptosis upon MCMV infection. Moreover, in vivo studies revealed that ZBP1-dependent virus detection played an important role in containing viral replication. More recently, Thapa and colleagues (Cell Host & Microbe 2016) convincingly showed that ZBP1 was also important for the recognition of Influenza A Virus leading to necroptosis induction in vitro and thereby containment of viral replication in vivo. In this study, it was furthermore demonstrated that ZBP1 directly bound to viral RNA. These studies thereby established ZBP1 as a bona fide PRR that binds RNA. This current manuscript now extends these findings to MCMV, by showing that MCMV-derived RNA transcripts are required to trigger ZBP1 activation leading to necroptosis.

An interesting observation that the authors make is that ZBP1 binds to endogenous RNA under conditions where it is overexpressed (Fig. 6). The authors suggest that endogenous RNA binding by ZBP1 might be involved in triggering necroptosis once caspases are inhibited (see Fig. 5), yet these studies were performed in cells expressing non-physiological amounts of ZBP1, whereas WT cells (with normal levels of ZBP1) don't show this phenotype. As such, the physiological relevance of these findings remains unclear.

*Altogether, the experiments are well performed and well described. However, in light of previously published work on this topic, this manuscript provides only little, incremental insight into the biology of ZBP1. As such, I do not find this manuscript suitable for publication at The EMBO Journal in its current form. I would suggest that the authors use their mouse model (ZBP1-*Zα1α2*mut) to study the role of ZBP1 in the context of perinatal lethality of RIPK1-deficient mice (see recent work by the labs of Dixit and Pasparakis). This way they could address whether ZBP1*

detects endogenous RNAs under physiological conditions (which is counteracted by RIPK1 under steady state conditions).

We thank the reviewer for his / her assessment of our manuscript and appreciate his / her positive comments on the quality of the performed experiments. We agree with the reviewer that our study builds on earlier and elegant work, particularly that published by Ed Mocarski's group (Upton et al., 2012). At the same time, we respectfully disagree with the comment on the novelty of our study. We would therefore like to highlight the importance of our findings:

- As pointed out by the reviewer, Thapa et al. proposed that ZBP1 binds to influenza A virus genomic RNA (Thapa et al., 2016). However, another study by Kanneganti and colleagues (Kuriakose et al., 2016) suggests that ZBP1 instead senses viral proteins. Therefore, whether ZBP1 is "a bona fide PRR that binds RNA" is controversial and remains to be established. We provide new evidence that ZBP1 is indeed a sensor for viral RNA, using several different techniques and well-performed experiments as pointed out by the reviewer.
- ZBP1 was initially described as a cytosolic B-DNA sensor that induces type I interferon and the name DAI (DNA-dependent activator of IFN-regulatory factors) was proposed for the protein (Takaoka et al., 2007). However, knock-out mice did not confirm this conclusion (Ishii et al., 2008) and we now know that cGAS is the key DNA sensor for interferon induction. We believe that our data now rigorously establish the "true" function of the molecule, i.e. RNA sensing leading to necroptosis.
- Nucleic acids can adopt the unusual Z-conformation. Despite the fact that this discovery was made many decades ago, biological functions of Z nucleic acids have remained elusive. Our data suggest an important physiological role of Z-RNA and thus address a long-standing and fundamental question in molecular biology.
- As mentioned by the reviewer, two recent papers published in *Nature* (Newton et al., 2016 and Lin et al., 2016) describe a role for ZBP1 in development and necroptosis in sterile conditions. An important open question is whether the function of ZBP1 in development is linked to sensing of nucleic acids. Our observation that ZBP1 can bind endogenous RNA supports this idea. We agree with the reviewer that our knock-in mouse will be an important tool to test this further. However, we believe that such studies are beyond the scope of this manuscript and will form the basis of future research. Importantly, we were advised by the editor not to embark on such studies as part of the revisions of this manuscript.

Minor point:

Without direct proof, I wouldn't state that ZBP1 detects Z-form RNA (at present, the authors have only shown that ZBP1 binds RNA, yet whether the bound RNA is of Z-form has not been shown). We agree with this comment and we would like to refer the reviewer to our response to reviewer #2 (point 1).

REFERENCES

- Daley-Bauer LP, Roback L, Crosby LN, McCormick AL, Feng Y, Kaiser WJ, Mocarski ES (2017) Mouse cytomegalovirus M36 and M45 death suppressors cooperate to prevent inflammation resulting from antiviral programmed cell death pathways. *Proceedings of the National Academy of Sciences of the United States of America* **114**: E2786-E2795
- Hall K, Cruz P, Tinoco I, Jr., Jovin TM, van de Sande JH (1984) 'Z-RNA'--a left-handed RNA double helix. *Nature* **311**: 584-586
- Hardin CC, Zarling DA, Puglisi JD, Trulson MO, Davis PW, Tinoco I, Jr. (1987) Stabilization of Z-RNA by chemical bromination and its recognition by anti-Z-DNA antibodies. *Biochemistry* **26**: 5191-5199
- Ishii KJ, Kawagoe T, Koyama S, Matsui K, Kumar H, Kawai T, Uematsu S, Takeuchi O, Takeshita F, Coban C, Akira S (2008) TANK-binding kinase-1 delineates innate and adaptive immune responses to DNA vaccines. *Nature* **451**: 725-729

Kaiser WJ, Upton JW, Long AB, Livingston-Rosanoff D, Daley-Bauer LP, Hakem R, Caspary T, Mocarski ES (2011) RIP3 mediates the embryonic lethality of caspase-8-deficient mice. *Nature* **471**: 368-372

Kuriakose T, Man SM, Malireddi RK, Karki R, Kesavardhana S, Place DE, Neale G, Vogel P, Kanneganti TD (2016) ZBP1/DAI is an innate sensor of influenza virus triggering the NLRP3 inflammasome and programmed cell death pathways. *Science immunology* **1**

Oberst A, Dillon CP, Weinlich R, McCormick LL, Fitzgerald P, Pop C, Hakem R, Salvesen GS, Green DR (2011) Catalytic activity of the caspase-8-FLIP(L) complex inhibits RIPK3-dependent necrosis. *Nature* **471**: 363-367

Popenda M, Milecki J, Adamiak RW (2004) High salt solution structure of a left-handed RNA double helix. *Nucleic acids research* **32**: 4044-4054

Ross WS, Hardin CC, Tinoco I, Jr., Rao SN, Pearlman DA, Kollman PA (1989) Effects of nucleotide bromination on the stabilities of Z-RNA and Z-DNA: a molecular mechanics/thermodynamic perturbation study. *Biopolymers* **28**: 1939-1957

Takaoka A, Wang Z, Choi MK, Yanai H, Negishi H, Ban T, Lu Y, Miyagishi M, Kodama T, Honda K, Ohba Y, Taniguchi T (2007) DAI (DLM-1/ZBP1) is a cytosolic DNA sensor and an activator of innate immune response. *Nature* **448**: 501-505

Thapa RJ, Ingram JP, Ragan KB, Nogusa S, Boyd DF, Benitez AA, Sridharan H, Kosoff R, Shubina M, Landsteiner VJ, Andrade M, Vogel P, Sigal LJ, tenOever BR, Thomas PG, Upton JW, Balachandran S (2016) DAI Senses Influenza A Virus Genomic RNA and Activates RIPK3-Dependent Cell Death. *Cell host & microbe* **20**: 674-681

Upton JW, Kaiser WJ, Mocarski ES (2010) Virus inhibition of RIP3-dependent necrosis. *Cell host & microbe* **7**: 302-313

Upton JW, Kaiser WJ, Mocarski ES (2012) DAI/ZBP1/DLM-1 complexes with RIP3 to mediate virus-induced programmed necrosis that is targeted by murine cytomegalovirus vIRA. *Cell host & microbe* **11**: 290-297

Wallach D, Kang TB, Dillon CP, Green DR (2016) Programmed necrosis in inflammation: Toward identification of the effector molecules. *Science* **352**: aaf2154

2nd Editorial Decision

24 May 2017

Thank you for submitting your revised manuscript to The EMBO Journal. I asked referee #2 to review the revised version and have now heard back from the referee.

As you can see below, the referee appreciates the introduced changes and is supportive of publication here. The only remaining comment is regarding the in vivo data that you present in the point-by-point response and that it would be good to include it in the paper. I agree with the referee on this point. How long will it take you to repeat the experiment? From our previous discussion regarding this I seem to recall that you have done the experiment twice - lets discuss further.

REFEREE REPORT

Referee #2:

I find that the authors have strengthened the work significantly, and generally addressed the points raised in a satisfactory manner. I do however find it a shame that the new in vivo data (shown in the PBP response) are not included in the manuscript. The editor should consider whether the data

should be included (even if the authors would need extra time to finalize the data).

2nd Revision - authors' response

04 June 2017

Response to referee #2:

I find that the authors have strengthened the work significantly, and generally addressed the points raised in a satisfactory manner.

The authors wish to thank reviewer 2 for critically evaluating our study again and for his/her positive comment.

*I do however find it a shame that the new *in vivo* data (shown in the PBP response) are not included in the manuscript. The editor should consider whether these data should be included (even if the authors would need extra time to finalize the data).*

We agree with the reviewer that these are important data that should be included. We have repeated the *in vivo* experiment in question and, after consulting the editor, now show these data in an additional panel (Fig 3G). Our results show that replication of MCMV-M45^{mutRHIM} is rescued in *Zbp1*^{Za1a2/Za1a2} animals. This observation extends our study into an *in vivo* setting and, we believe, strengthens our manuscript further.

For convenience, changes to the text are highlighted in blue. Anne Bridgeman from the Rehwinkel group is now a co-author as she helped with the mouse experiment. Additionally, we have included Katherine B. Ragan and Jason W. Upton (Austin, Texas) as co-authors; they provided MCMV for some of the new *in vivo* experiments.

Corresponding Author Name: Jan Rehwinkel

Journal Submitted to: The EMBO Journal

Manuscript Number: EMBOJ-2017-96476